# Exploring the past and forecasting the future of malaria in selected Nigerian states: A time series modelling approach using wavelet and SARIMA

Emmanuel Afolabi Bakare[1,2]*, Oluwaseun Akinlo Mogbojuri[1,2,3],
Dolapo Ayomide Bakare[1,4], Oluwakemi Janet Odewusi[1,2]

1 International Centre for Applied Mathematical Modelling and Data Analytics, Federal University Oye-Ekiti, Ekiti State, Nigeria, 2 Department of Mathematics, Federal University Oye-Ekiti, Ekiti State, Nigeria, 3 Department of Mathematical Sciences, Adekunle Ajasin University, Akungba-Akoko, Ondo State, Nigeria, 4 Department of Animal Production and Health, Federal University Oye-Ekiti, Ekiti State, Nigeria

☺ These authors contributed equally to this work.
* emmanuel.bakare@fuoye.edu.ng

## Abstract

Malaria remains a life-threatening disease and poses a significant economic burden in sub-Saharan Africa, with Nigeria accounting for the highest global morbidity and mortality. Despite increased preventive and control interventions implemented by the National Malaria Elimination Programme over the years, malaria transmission in Nigeria continues to exhibit spatial and temporal dynamics making elimination difficult to achieve. Understanding malaria seasonal patterns and synchrony between cases in different transmission settings in Nigeria, and forecasting future outbreaks is therefore critical for guiding public health policies. Time series modelling plays a crucial role in extracting meaningful insights, revealing patterns and forecasting infectious disease outbreaks. In this study, we employed two time series approaches-wavelet analysis and the Seasonal Auto-Regressive Integrated Moving Average (SARIMA) model-to analyse confirmed uncomplicated malaria case data of Nasarawa, Kwara, and Zamfara States, Nigeria, over a 10-year period (January 2015 to December 2024). Wavelet analysis decomposed the time series data into their constituent time-frequency components, enabling multi-resolution analysis of the data and revealing hidden patterns. The SARIMA model, on the other hand, was used to forecast malaria cases in each of the three states for the next two years (January 2025 to December 2026). Univariate wavelet analysis showed that malaria incidence in Nasarawa followed a very strong annual cycle throughout almost the entire study period, with seasonal peaks occurring mostly in September 70% of the time. In Kwara State, short periods of semi-annual, annual, and multi-annual cycles were detected, with annual peaks mostly appearing in August for about half of the study period. In Zamfara, a moderate yearly cycle was observed between 2015 and June 2020 and again from 2022 to 2023, with seasonal peaks occurring consistently in September, except in 2024, when the peak shifted to August. Bivariate wavelet

**Data availability statement:** All relevant data for this study are publicly available from the figshare repository (https://doi.org/10.6084/m9.figshare.30716729).

**Funding:** This study was funded by a grant from the Bill & Melinda Gates Foundation (BMGF), grant number INV-047051 (Grand Challenges in Global Health). The funders had no role or influence on the design and interpretation of the data collected, as well as in writing the manuscript.

**Competing interests:** The authors have declared that no competing interests exist.

analysis showed that Nasarawa and Zamfara had an almost perfectly synchronized seasonal malaria pattern for most of the study period. For both the Nasarawa-Kwara and Zamfara-Kwara pairs, synchrony was present only between mid-2018 and early-2020; outside this period, Kwara's malaria peaks occurred earlier in the season in both cases. For the period 2025–2026, the SARIMA model forecasted average monthly malaria cases of 51,482 (95% CI: 23,899-116,073) in Nasarawa; 20,850 (95% CI: 3,381-202,511) in Kwara; and 67,463 (95% CI: 27,317-171,418) in Zamfara. These findings emphasize the need for state-specific malaria interventions to capture the variability observed in Kwara, and regionally coordinated control measures for Nasarawa and Zamfara where very strong synchrony exists. They also provide useful guide for policymakers on optimally timing interventions in line with observed seasonal patterns.

## Introduction

Malaria is an infectious disease caused by parasites of the genus *Plasmodium*, transmitted by the bite of infected female Anopheles mosquitoes [1]. In humans, it is caused by *Plasmodium falciparum*, *Plasmodium malariae*, *Plasmodium ovale*, *Plasmodium knowlesi* and *Plasmodium vivax* [2]. Of these, *P. falciparum* is the most common cause of infection in sub-Saharan Africa and South-East Asia, and is responsible for 80% of all malaria cases and 90% of deaths [3]. *Plasmodium vivax* typically causes milder infections than *P. falciparum* but has a much greater geographical distribution [4]. The clinical symptoms of malaria are largely a result of the replication of asexual stages in human blood, but transmission to mosquitoes is only achieved through the development of sexual stages, termed gametocytes [5,6].

Malaria is a major public health concern in many countries worldwide [7]. Although completely preventable, malaria has a high level of mortality and is the most prevalent parasitic disease in the world [8]. The disease has a significant impact on social, economic and health aspects, which continues to be a major public health challenge [9]. According to the World Health Organization [2], an estimated 263 million cases of malaria and 597,000 malaria deaths occurred globally in 83 countries in 2023. The burden of malaria worldwide is disproportionately high in sub-Saharan Africa with 94% of malaria cases (246 million) and 95% (569,000) of malaria deaths occurring in the region in 2023 [2]. Children under the age of five are the most affected by malaria with approximately 76% of all malaria deaths in sub-Saharan Africa occurring in that age group [2].

While malaria transmission in Southern Nigeria is all year-round, transmission in the Northern part of the country is highly seasonal, with 97% of the entire population at risk of being infected with malaria [10,11]. According to the 2022 World Malaria Report [10], Nigeria accounts for the highest percentage of the global malaria burden compared to any other country, with 27% of the global estimated malaria cases and 31% of the estimated deaths, as well as an estimated 55% of malaria cases in West Africa in 2022 [10].

The National Malaria Elimination Programme (NMEP) has implemented four National Malaria Strategic Plans (NMSPs) since 2001 to combat the problem of malaria in Nigeria [12]. It is presently in its final year of its fifth plan, which runs from 2021 to 2025 [12]. To reach its goals of a parasite prevalence of less than 10% and a reduction in malaria-related mortality to fewer than 50 deaths per 1,000 live births by 2025, the NMEP has put in place a number of preventive and control interventions [12]. These interventions include seasonal malaria chemoprevention (SMC), intermittent preventative treatment (IPT), larva source management (LSM), case management (CM), long-lasting insecticidal nets (LLINs) etc [12].

In the past few years, time series modelling has become increasingly useful in extracting meaningful insights, revealing patterns and forecasting disease outbreaks [13]. Wavelet analysis, an emerging powerful time series tool for identifying hidden patterns and providing valuable insights into underlying processes has found applications in several areas of endeavour including finance and economics [14–16], engineering and ecology [17,18], geophysics [19,20], climatology [21,22] and epidemiology [23,24].

Wavelet analysis has a great advantage of being free from the stationarity assumption, a characteristic property of most epidemiological time series [23]. Although wavelet analysis does not provide any information about underlying transmission mechanism of infectious diseases such as malaria, it does give useful insights about the nature of the underlying epidemiological processes [18,23,25]. The insights obtained from wavelet analysis lay the groundwork in incorporating explicit mechanisms for future modelling approaches [23]. Thus, wavelet analysis can be employed as a first step before any modelling process, to investigate the complexity of the observed epidemiological time series [23].

Several researches have been carried out using wavelet analysis to explore the transient dynamics of infectious diseases. The authors in [26] investigated the epidemiological characteristics of influenza virus and its association with climatic factors in Jinan, China using wavelet analysis. By using data from three influenza sentinel hospitals in Jinan, China as well as climatic data from 2013 to 2016, they showed that influenza dynamics was characterised by annual cycle, with remarkable peaks from December to February of the study period. Their findings revealed that during the influenza epidemic season in Jinan, China, climatic factors are significantly correlated with influenza seasonality.

The seasonal patterns of influenza and their association with meteorological as well as air pollution factors were investigated by the authors in [27] across six regions of Thailand. They used wavelet analysis to analyse meteorological, air pollution and influenza incidence data from 2009 to 2019. Their study revealed inconsistent biannual influenza patterns throughout the period of the study. Wavelet analysis has also been applied in investigating several other infectious diseases such as COVID-19 [28–31], dengue [25,32] and mpox [33].

Forecasting models such as the Auto-Regressive Integrated Moving Average (ARIMA) and Seasonal ARIMA (SARIMA) models have played crucial roles in helping public health policy makers plan and manage disease outbreaks effectively. A substantial amount of literature has been devoted to forecasting outbreaks of infectious diseases (see, e.g., [33–42]).

In spite of the numerous studies done in applying wavelet analysis to investigate the dynamics of infectious diseases, very little research has been devoted to malaria. The interactions between climatic factors and malaria interventions across the three different climatic zones in Burkina Faso were assessed using wavelet analysis in [43]. In Sri Lanka wavelet analysis was used to identify patterns of malaria transmission and its association with environmental factors between the time period from 1990–2005 [44]. In [45], the spatial and temporal patterns of *P. falciparum* and *P. vivax* malaria were identified using wavelet techniques. Furthermore, in [46], wavelet approaches were employed to investigate the variation of malaria dynamics and its relationship to climate in western Kenya. Few other studies have explored malaria dynamics using wavelet techniques [47–50]. To the best of our knowledge, only [34] has made an attempt to assess the variability of malaria in pregnancy in Nigeria using wavelet analysis. A notable gap in the reviewed studies is the paucity of research dedicated to understanding malaria trends and seasonal patterns within the Nigeria context. This is particularly worrisome given the country's foremost contribution to the global malaria burden. Furthermore, the effectiveness of interventions such as SMC, introduced to tailor malaria control strategies to geographic areas with seasonal transmission, is increasingly challenged by the impacts of climate change [51].

This study therefore aims to address this gap in knowledge by extracting meaningful information, uncovering hidden seasonal patterns in malaria dynamics in Nasarawa, Kwara, and Zamfara States, Nigeria, and forecasting the trends and burden of malaria cases in these states over the next two years. This is done by applying wavelet analysis and Seasonal Auto-Regressive Integrated Moving Average (SARIMA) model.

The remaining part of the paper is structured as follows. The Materials and methods section provides a detailed account of the materials and methods used in this research, including the study area, data description, and the modelling approaches employed. The Results section presents the findings obtained from the analysis, followed by the Discussion section where the results are interpreted and their implications are explained. Finally, the Conclusion section presents the concluding remarks and summarizes the main findings of the study.

## Materials and methods

### Study area

This study was carried out using malaria incidence data from three Northern Nigerian states: Nasarawa and Kwara in the North Central region, and Zamfara in the North West region. Fig 1 presents the map of Nigeria showing the three states under consideration. Geographically, Nasarawa, Kwara, and Zamfara States are located at approximately 8.55°N, 9.08°N, and 12.07°N latitude, and 7.71°E, 4.47°E, and 6.05°E longitude, respectively. Their estimated populations in 2024 were 2,371,702 for Nasarawa, 3,012,797 for Kwara, and 4,187,897 for Zamfara [52]. These states were chosen in order to reflect varying malaria prevalence rates in Northern Nigeria. According to the 2021 Nigeria malaria indicator survey [53], by microscopy, Zamfara has a high malaria prevalence of 37%, Nasarawa has a medium malaria prevalence of 15% while Kwara has a relatively low malaria prevalence at 6%.

Nasarawa State experiences two distinct seasons: the dry season (November to February) and the rainy season from March to October with peaks in August [54]. The State contributed an estimated 1.2% of Nigeria's 68 million malaria cases

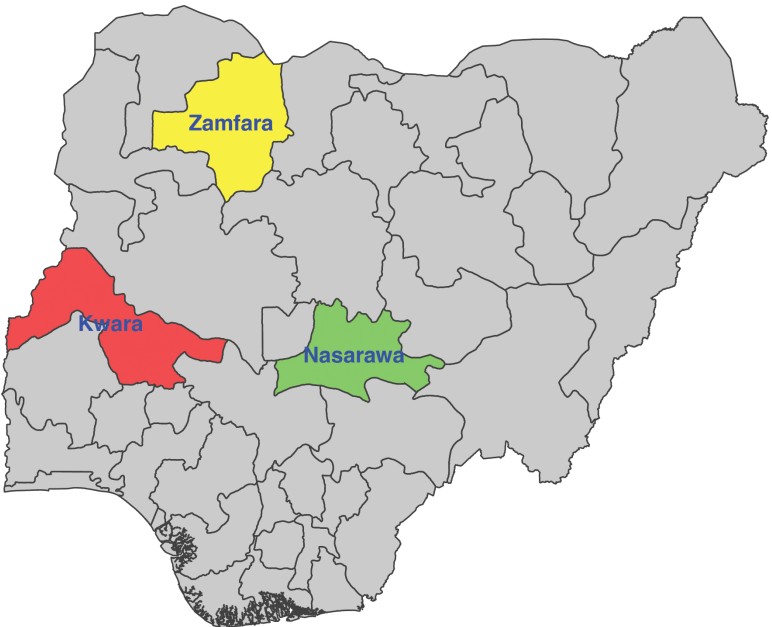

**Fig 1. Map of Nigeria showing Nasarawa, Kwara and Zamfara States.** This figure was created by the authors using R programming software (R version 4.4.3). Available at (https://www.R-project.org/). The shapefile was obtained from openly available country boundary data (Administrative boundary data) from Global Administrative Areas (GADM) (https://gadm.org/license.html), which permits academic use.

in 2021 [54]. There are two different seasons in Kwara State: the rainy season from May to September when it peaks and the dry season (October to April). An estimated 1.4% of Nigeria's 68 million malaria cases in 2021 were attributable to the State [54]. Zamfara State also experiences two distinct seasons: the rainy season (May–September, with peaks in August) and the dry season (October–April). In 2021, it accounted for an estimated 2.5% of Nigeria's 68 million malaria cases [54].

The different transmission settings represented by each of the three states provides a strong justification for comparing malaria dynamics within Northern Nigeria.

## Data description and preprocessing

The dataset used in the SARIMA modelling comprise monthly reported malaria cases of each of the three states under consideration, covering a period of 120 months – from January 2015 to December 2024. The dataset includes 120 observations, each representing a month, and 14 variables including number of confirmed uncomplicated malaria, number of severe malaria etc. The processed data used for the wavelet analysis includes variables such as the projected population of each of the states from 2015 to 2024 obtained from [52] and their coordinates. The monthly reported malaria cases data were obtained from the National Malaria Data Repository (NMDR (https://nmdrnigeria.ng/dhis-webcommons/security/login.action). All the dataset used in the study were sourced in a fully de-identified form between 18 April 2025 and 10 May 2025, and can be accessed via the figshare repository (https://doi.org/10.6084/m9.figshare.30716729). No personal identifiers were collected.

The monthly reported confirmed uncomplicated malaria cases for each of the three states were preprocessed in order to prepare the data for time series analysis. The dataset was examined for missing values and was found to contain none. For the SARIMA model, the absolute case counts for each of the three states were utilized. The case counts were subsequently log transformed in order to stabilise the variance and reduce the influence of outliers. For wavelet analysis, incidence rates per 1000 population were calculated for each of the states. A log transformation was then applied to the incidence rates to stabilize variance and make the time series more normally distributed. Detrending was applied by regressing the log-incidence series on time and retaining the residuals. This is important for wavelet analysis as long-term trends could obscure underlying seasonal and cyclic patterns. The detrended series were subsequently standardised to zero mean and unit variance so as to ensure that the different time series of each state have comparable scales when performing cross-wavelet analysis.

## Wavelet analysis

The traditional Fourier analysis has been used for a very long time to decompose time series into its sinusoidal components of varying frequencies [24,55]. Despite its great usefulness in helping to determine the several spectral components in a time series, Fourier analysis is unable to provide the time information when these frequencies occur [23,55]. Moreover, Fourier analysis is suitable for stationary time series – that is, time series whose statistical properties are constant in time [18]. However, most epidemiological time series are highly non-stationary. Several solutions have been developed in order to circumvent the limitations of Fourier analysis [23]. One of such is the short time or windowed Fourier transform introduced in 1964 by Dennis Gabor to quantify the time-frequency component of time series [14,23]. This method is however, ineffective due to the same frequency resolution across all frequencies. The wavelet analysis has been proposed as a more efficient approach to these limitations [14,18].

The wavelet transform has the advantage of being able to perform natural local analysis of a time series by adapting the length of the wavelet function to the signal's frequency content. it stretches into a long wavelet to capture low-frequency structures and compresses into a short wavelet to capture high-frequency features [14]. This allows the wavelet transform to decompose a time series using functions that are narrow in time for high-frequency features and wider for analyzing long-term, low-frequency variations [18]. In other words, the wavelet transform helps to capture abrupt changes in the time series by using very short functions (narrow windows) for high-frequency components, while also isolating slow

and persistent movements through very long functions (wide windows). This adaptive feature makes wavelet analysis ideal for studying signals with both rapid fluctuations and long-term trends [14,18].

**The continuous wavelet transform.** A wave is described as an oscillating function of time or space, such as a sinusoid. Wavelets are "small waves" with their energy concentrated in time, making them effective tools for analyzing transient, non-stationary, or time-varying phenomena [56]. Wavelets are defined by the wavelet function $\psi(t)$ typically referred to as the "mother wavelet". Given a "mother wavelet" $\psi(t)$, a family $\psi_{a,\tau}(t)$ of wavelets called "daughter wavelets" can be obtained by scaling $\psi(t)$ by $a$ and translating it by $\tau$ [14]:

$$\psi_{a,\tau}(t) = \frac{1}{\sqrt{|a|}}\psi\left(\frac{t-\tau}{a}\right), \quad a, \tau \in \mathbb{R}, a \neq 0.$$

(1)

Here, the parameter $a$ is the dilation or scale which represents how stretched ($|a| > 1$) or compressed ($|a| < 1$) the wavelet is. At larger scales, the wavelet becomes wider and covers a longer portion of the time series, causing details to become blurred. $\tau$ represents the translation or time shift which refers to the position of the wavelet as it moves horizontally along the time axis as shown in Fig 2. By shifting the wavelet across the time series, the transform captures how the series' features vary at different points in time. The term $\frac{1}{\sqrt{|a|}}$ helps to normalize the wavelets, ensuring they have a uniform variance of 1 across all scales.

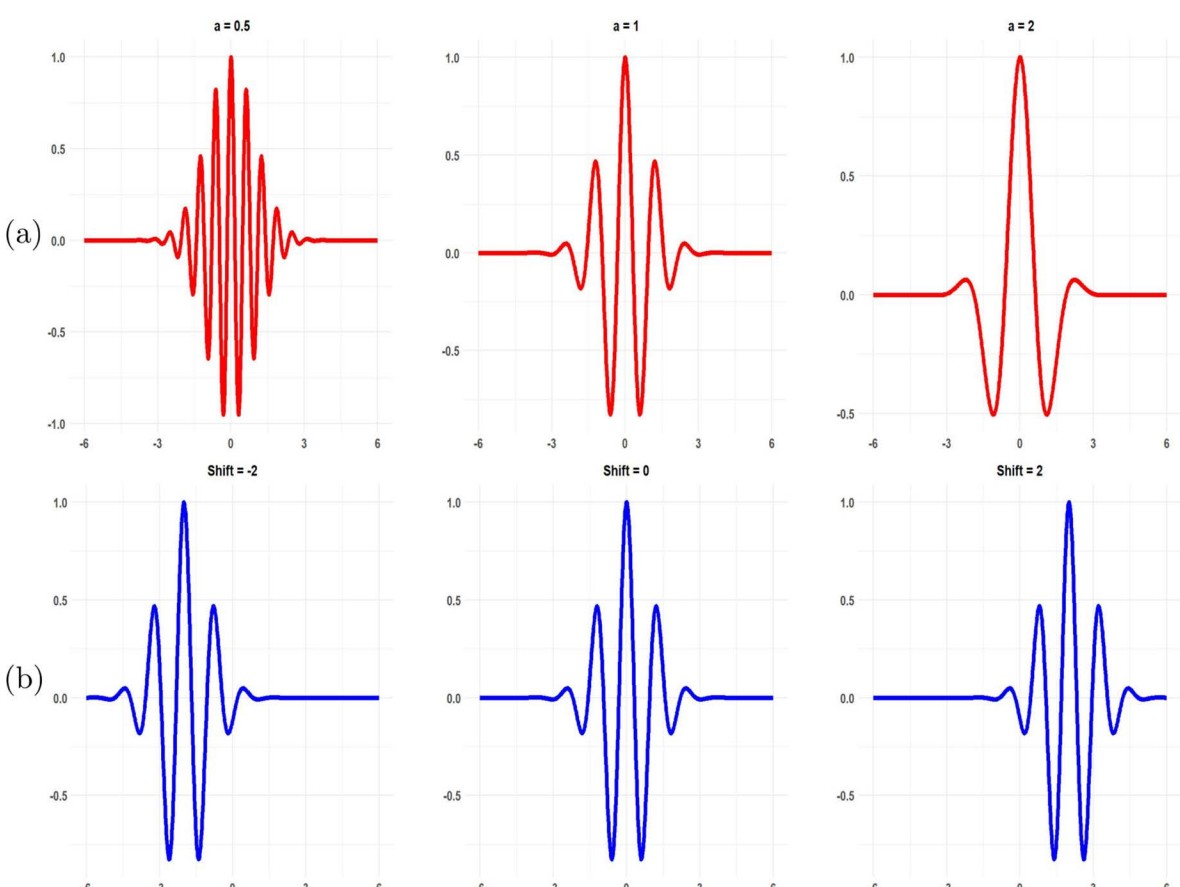

**Fig 2. Morlet wavelets at different: (a) scales and (b) translations (shifts) (Source: [33]).**

The continuous wavelet transform of a signal or time series $x(t)$ with respect to a mother wavelet $\psi(t)$ is defined by [18,23]

$$W_x(a, \tau) = \frac{1}{\sqrt{|a|}} \int_{-\infty}^{\infty} x(t)\psi^* \left( \frac{t - \tau}{a} \right) dt,$$

(2)

where $\psi^*(t)$ denotes the complex conjugate form of $\psi(t)$. The wavelet coefficients $W_x(a, \tau)$ denote the contribution of the scales $a$ at various time positions $\tau$. The wavelet transform (2) can therefore be viewed as a cross-correlation of a time-series $x(t)$ with a family of wavelets $\psi\left(\frac{t-\tau}{a}\right)$ of different scales $a$, at different time positions $\tau$. Commonly used wavelet functions in the literature include the Morlet, Mexican Hat, Haar, and Daubechies wavelets. Among these, the Morlet wavelet which is widely popular for its good time-frequency localisation is employed in this study. It is defined as [14,23].

$$\psi(t) = \pi^{-\frac{1}{4}} e^{i\omega_0 t} e^{-\frac{t^2}{2}}$$

(3)

where $\omega_0$ is the central angular frequency of the wavelet.

**Wavelet power spectrum.** The wavelet power spectrum shows how the variance (energy) of a time series is distributed across different frequencies and time. At scale $f$ and time $\tau$, the wavelet power spectrum is given by [18]

$$S_x(f, \tau) = ||W_x(f, \tau)||^2.$$

(4)

The global wavelet power spectrum or the average power spectrum is defined as the averaged variance contained in all wavelet coefficients of the same frequency $f$ and is given by [23].

$$\overline{S}_x(f) = \frac{\sigma_x^2}{T} \int_0^T ||W_x(f, \tau)||^2 \, d\tau,$$

(5)

where $\sigma_x^2$ and $T$ are the variance and duration of the time series respectively. The global wavelet spectrum provides an unbiased and reliable estimation of the actual power spectrum of a time series, typically exhibiting analogous characteristics and form to the corresponding Fourier spectrum [20,23].

**Wavelet coherency.** Coherency is used to assess the association between two time series $x(t)$ and $y(t)$. The coherence function provides a direct measure of how closely the spectra of the two time series are correlated [18]. The wavelet cross-spectrum and the wavelet coherence can be used to quantify the relationship between two non-stationary time series at different frequencies and how they evolve over time [14,18,57]. The wavelet cross-spectrum of two time series $x(t)$ and $y(t)$ is defined as [18,23]

$$W_{x,y}(f, \tau) = W_x(f, \tau)W_y^*(f, \tau)$$

(6)

where $W_x(f, \tau)$ and $W_y(f, \tau)$ are the wavelet transforms of $x(t)$ and $y(t)$ respectively, and * represents the complex conjugate. While the wavelet power spectrum represents the local variance of a single time series, the wavelet cross-spectrum of two time series reflects their local covariance at each scale or frequency. In otherwords, a wavelet cross-spectrum provides a quantified indication of how similar the power of the two time series is across time and frequency [14].

Wavelet coherency is defined as the cross-spectrum normalized by the spectrum of each time series and is given by [14,23]:

$$R_{x,y}(f, \tau) = \frac{\|\langle W_{x,y}(f, \tau)\rangle\|^2}{\|\langle W_{x,x}(f, \tau)\rangle\|^{1/2} \cdot \|\langle W_{y,y}(f, \tau)\rangle\|^{1/2}},$$

(7)

where "$\langle\rangle$" represents a smoothing operator both in time and frequency. The wavelet coherency can be viewed as a localized correlation coefficient in time-frequency space and ranges between 0 and 1 [14,20,58].

**Phase difference.** The phase of a time series $x(t)$ represents its position within a pseudo-cycle and is typically measured in radians, ranging from $-\pi$ to $\pi$ [23]. Given two time series $x(t)$ and $y(t)$, the phase difference provides information about the delays of the oscillations between them. That is, it provides information about the time between a peak in $x(t)$, the leading time series, and the nearest subsequent peak in $y(t)$, the lagging series [14,18]. Thus, it characterizes the phase relationship between the two time series. The phase difference is defined as [18,23]

$$\phi_{x,y}(f, \tau) = \phi_x(f, \tau) - \phi_y(f, \tau) = \tan^{-1} \frac{\Im\left(\langle W_{x,y}(f, \tau)\rangle\right)}{\Re\left(\langle W_{x,y}(f, \tau)\rangle\right)},$$

(8)

where $\Im$ and $\Re$ are the imaginary and real parts respectively. A phase difference of 0 radians implies that the two time series are synchronized and oscillate in unison at the specified frequency. That is, their peaks and troughs are in alignment. A phase difference of $\pi$ radians or $-\pi$ radians indicates that the two time series are in a perfect anti-phase relationship (i.e., one peaks when the other troughs). If $\phi_{x,y}$ lies between 0 and $\frac{\pi}{2}$, the two series are in phase, but $x(t)$ leads $y(t)$. If $\phi_{x,y}$ lies between $-\frac{\pi}{2}$ and 0, they are also in phase, but $y(t)$ leads $x(t)$. When $\phi_{x,y}$ falls between $\frac{\pi}{2}$ and $\pi$, $y(t)$ is leading but out of phase with $x(t)$. Conversely, if $\phi_{x,y}$ is between $-\pi$ and $-\frac{\pi}{2}$, it is $x(t)$ that leads and the series are out of phase [14,58].

## ARIMA model

The ARIMA model is a commonly used statistical method for analysing and forecasting time series data [33,39,40]. In 1976, Box and Jenkins presented the ARIMA model, usually denoted by ARIMA ($p$, $d$, $q$), as a tool for predicting non-seasonal time-series data [59].

In this model, $p$ stands for the order of the autoregressive (AR) component. The amount of non-seasonal differencing needed to reach stationarity is indicated by $d$, and the order of the moving average (MA) component is denoted by $q$ [60]. In order to predict the variable of interest, AR simply involves regressing the variable against itself. It compares the one-time period's pattern to its earlier periods. MA is a regression-like model that forecasts a variable at a later time-step by using the forecast errors from a prior time-step [59]. The generalized equations of the $p^{th}$ order autoregressive (AR) model and the $q^{th}$ order moving average (MA) model are given by equations (9) and (10) respectively.

$$y_t = C + \phi_1 y_{t-1} + \phi_2 y_{t-2} + \cdots + \phi_p y_{t-p} + \varepsilon_t$$

(9)

$$y_t = C + \varepsilon_t + \theta_1 \varepsilon_{t-1} + \theta_2 \varepsilon_{t-2} + \cdots + \theta_q \varepsilon_{t-q}$$

(10)

ARIMA models are built by combining the AR model (9), integration (I), and the MA model (10). The integration component refers to the differencing process applied to make a non-stationary series stationary; forecasting involves reversing this differencing [59]. The ARIMA model is expressed as:

$$y_t = C + \phi_1 y_{t-1} + \phi_2 y_{t-2} + \cdots + \phi_n y_{t-n} + \theta_1 \varepsilon_{t-1} + \theta_2 \varepsilon_{t-2} + \cdots + \theta_q \varepsilon_{t-q} + \varepsilon_t.$$

(11)

Here, $C$ denotes the intercept; $\phi_i$ ($i = 1, 2, \ldots, p$) are the parameters of the AR component; $\theta_i$ ($i = 1, 2, \ldots, q$) are the parameters of the MA component; $y_t$ is the value of the time series at time $t$; $y_{t-1}, y_{t-2}, \ldots, y_{t-p}$ represent the previous values of the series; and $\varepsilon_t$ denotes the random error or residual term at time $t$ [59].

## SARIMA model

Time-series data frequently show seasonal patterns in many real-world datasets that are difficult for a basic ARIMA model to accurately represent [61]. The Seasonal Autoregressive Integrated Moving Average (SARIMA) model is employed to get around this restriction. In order to simulate both short-term dependencies and recurrent seasonal fluctuations, SARIMA expands the traditional ARIMA framework by adding extra seasonal components [61]. SARIMA $(p, d, q)(P, D, Q)s$ is the standard expression for the SARIMA model, where $p$, $d$ and $q$ stand for the non-seasonal autoregressive, differencing, and moving average orders, respectively while $P$, $D$ and $Q$ represents the corresponding seasonal parts. The seasonal terms ($P$, $D$, and $Q$) function across a specific seasonal cycle, indicated by s, which is the number of time steps in a full seasonal period [59]. The SARIMA model is expressed mathematically as:

$$\Phi_P(B^m)\,\phi_p(B)\,(1-B^m)^D\,(1-B)^d\,y_t = \Theta_Q(B^m)\,\theta_q(B)\,w_t \tag{12}$$

The non-stationary time-series is represented by $Y_t$ in this context, while the Gaussian white noise process is denoted by $w_t$. The non-seasonal autoregressive and moving average polynomials are expressed as $\phi(B)$ and $\theta(B)$, respectively. Seasonal behaviour is captured by the seasonal autoregressive polynomial $\Phi_P(B^m)$ and the seasonal moving average polynomial $\Theta_Q(B^m)$.

Seasonal trends are removed using the differencing parameter $D$, which can take values such as 1 or 2 depending on the characteristics of the data; however, stationarity is often achieved with $D = 1$. Lagged terms in the series are expressed using the operator $B$, known as the backshift operator, which is defined as shown in Eq 13.

$$B^k Y_t = Y_{t-k} \tag{13}$$

## Analytical tools and model evaluation

A combination of statistical and error-based measures was used in this work to select models and assess their performance for time-series forecasting. Despite being frequently used to identify the proper parameters in conventional ARIMA modelling, tools like the Auto-Correlation Function (ACF) and Partial Auto-Correlation Function (PACF) were not utilized in this study. Rather, the Auto ARIMA function in R, which used the Akaike Information Criterion (AIC) to evaluate parameter combinations iteratively and select the best-fitting ARIMA models was used. Following model selection, three common error metrics, Mean Absolute Error (MAE), Mean Squared Error (MSE), and Root Mean Squared Error (RMSE), were used to evaluate prediction accuracy. Because they quantify the average size of forecasting mistakes, these measures offer a thorough assessment of model performance.

**Model evaluation metrics.** MAE, RMSE, and MAPE are commonly used to evaluate the accuracy of forecasting models. The degree to which the actual value of an observation differs from its prediction is known as the absolute error. A measure of the magnitude of errors for a collection of predictions and observations is obtained by taking the mean absolute errors for the group. This is given by:

$$MAE = \frac{1}{n}\sum_{i=1}^{n}\left|y_i - \hat{y}_i\right| \tag{14}$$

The RMSE measures the average of the squared difference between a statistical model's predicted values and the actual values. Mathematically, it is the standard deviation of the residuals. It is given by:

$$RMSE = \sqrt{\frac{1}{n}\sum_{i=1}^{n}(y_i - \hat{y}_i)^2} \tag{15}$$

The MAPE represents the absolute error between the actual and anticipated values as a percentage of the actual values. It is expressed mathematically as:

$$\text{MAPE} = \frac{1}{n} \sum_{i=1}^{n} \left| \frac{y_i - \hat{y}_i}{y_i} \right|$$

(16)

## Analysis of data

In this study, the biwavelet package in R was used to perform both univariate and bivariate wavelet analyses while the CircStats R package provides functions for circular statistics, which are useful for analyzing phase angles. The analysis was carried out on the normalised, detrended log-incidence series using a Morlet wavelet, with a maximum period of 48 months. Wavelet power spectrum and cross-wavelet power spectrum were generated for the series assuming a 5% significance level. The original time series was reconstructed by filtering specific periodicity bands from the wavelet transform. Specifically, the reconstruction was done by filtering the periods between 9 and 15 months, corresponding to the dominant seasonal cycle of malaria in the selected states.

Statistical significance for the wavelet power spectrum was determined against a red noise background (first order autoregressive, AR(1)) using a Chi-square ($\chi^2$) test with two degrees of freedom ($p<0.05$), following the theoretical framework established in [57]. For the cross-wavelet spectrum between the time series of any two states, significance levels were estimated using Monte Carlo methods with 600 random iterations. The null hypothesis was based on the first-order autoregressive (AR1) process of each input time series, with the 95% confidence interval derived from the simulated distribution. For all analyses, a cone of influence (COI) was applied to avoid edge effect and features within the COI are interpreted with caution.

## Results

### Univariate analysis of malaria incidence

The wavelet power spectra of malaria incidence in each of the states are presented in Fig 3. The colour code for power ranges from deep blue (low power) to deep red (high power). The cone of influence (COI), which shows the region where edge effects are significant, is described using a white triangular line. For Nasarawa State, the deep red area between the 9–15 month periodic band indicates a strong annual cycle in malaria cases almost throughout the duration of the time series (Fig 3a). A second, but weak power band is visible around the 24-month period, particularly around March 2017 to March 2022 (Fig 3a). The global wavelet power spectrum shows a prominent peak around 9–15 months, suggesting a strong annual cycle in malaria incidence in Nasarawa (Fig 3b).

A deep red patch is observed in the local wavelet power spectrum of Kwara State around the 30–42 periodic band between January 2015–August 2017, indicating a dominant 2.5-3.5 year cycle (Fig 3c). However, this feature appear within the COI. Consequently, this should be interpreted cautiously. Red patches were also observed around the 6 month and 12 month periodic bands. These suggest semi-annual and annual periodic patterns which were detected from the beginning of the series until around February 2016 and February 2017, respectively, although most of these patterns fall largely within the COI and so should be treated with caution. The power of these two signals decrease up till the end of the observed period. The global wavelet power spectrum confirms the presence of both short-term cycles and long-term cycles (Fig 3d).

The wavelet power spectrum of malaria incidence for Zamfara State (Fig 3e) reveals moderate annual cycle around the 12 month periodic band from the start of the series to around late 2019 and also towards the end of the series around January 2022 to December 2023 although some portions fall within the COI. A much more dominant multi-annual cycle (3-3.5 year cycle) was observed at the 36−42 period band with a significant portion falling within the COI, and so great caution is required in interpreting these results. Fig 3f confirms the presence of a peak around 12 months suggesting an annual cycle and a much significant peak around 36−42 months indicating the presence of a 3-3.5 year cycle.

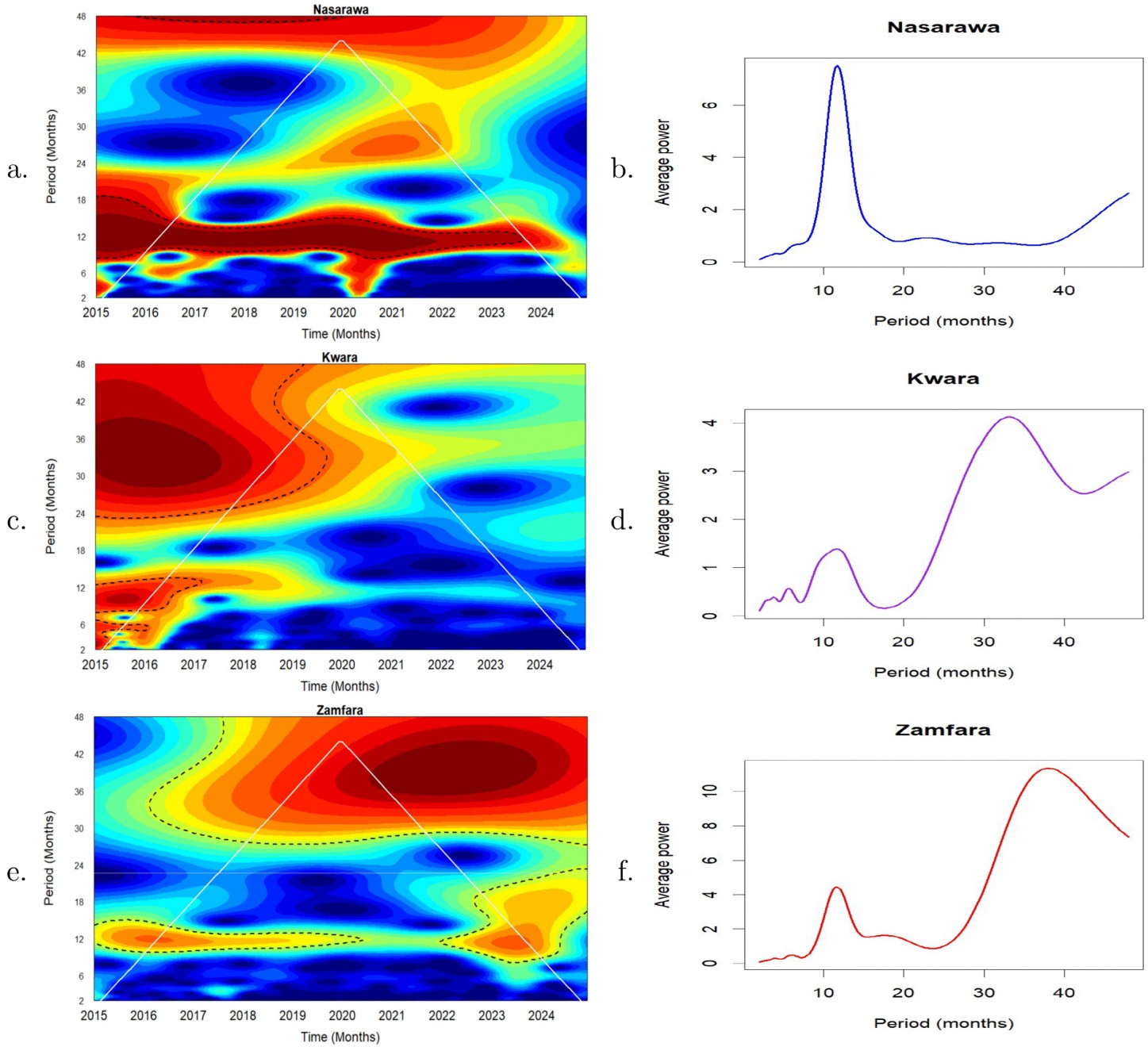

**Fig 3. Wavelet analysis of the transient relationship between malaria incidence in Nasarawa, Kwara and Zamfara from January 2015 to December 2024.** Left panel: Local wavelet power spectrum of malaria incidence in **a.** Nasarawa **c.** Kwara and **e.** Zamfara States. Right panel: Global wavelet power spectrum of malaria incidence in **b.** Nasarawa **d.** Kwara and **f.** Zamfara States. The black dashed contours 5% significance levels against red noise and the white line indicates the cone of influence that delimits the region not influenced by edge effects.

The wavelet transform was used to reproduce the normalised original malaria time series of each of the three states. The reconstruction was done based on the annual cycle (Fig 4a, c, e). The phase angles of the annual cycle for each of the three states are presented in Fig 4b, d, f. The phase angles for each state provide valuable information about the

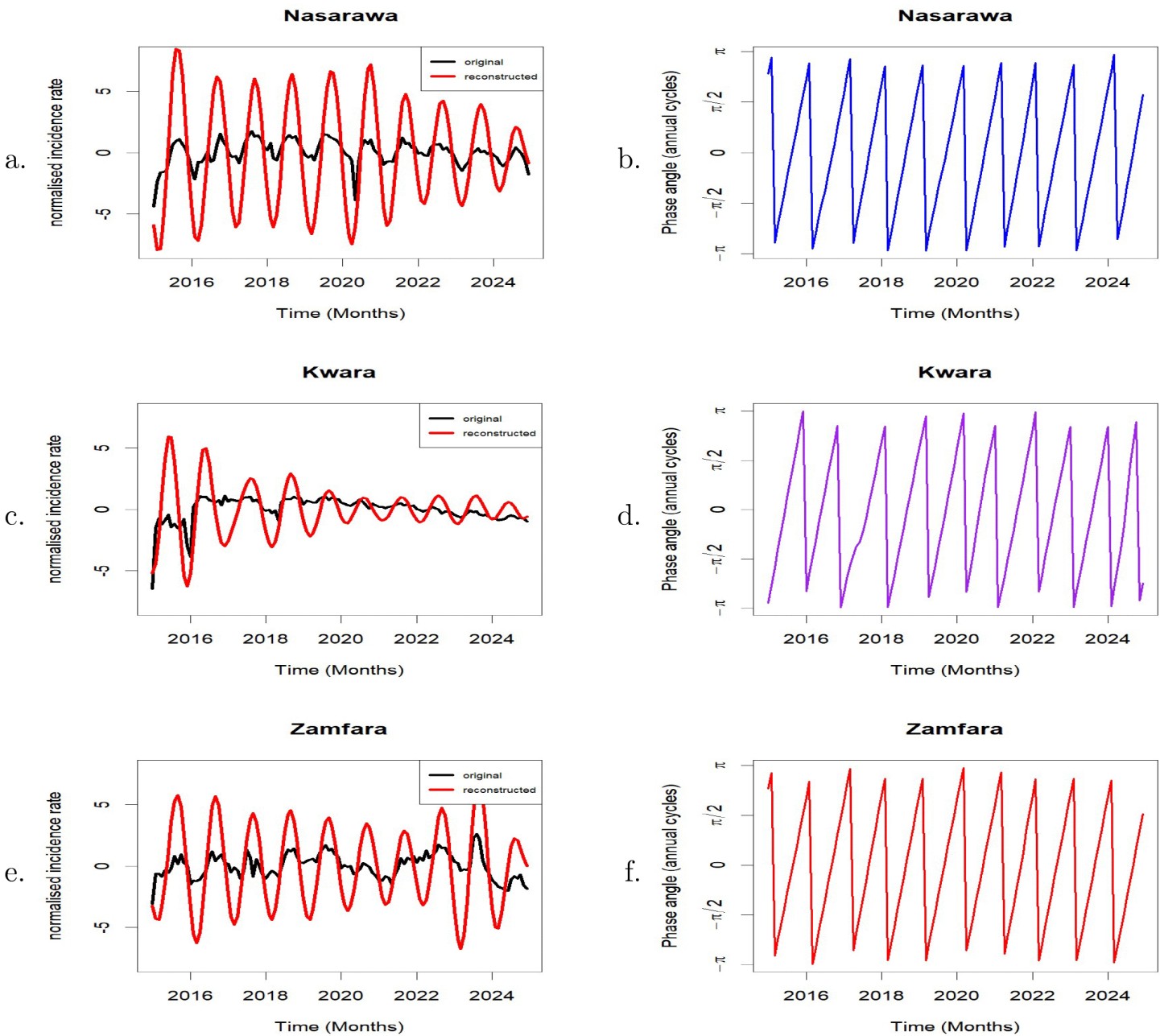

**Fig 4. Left panel (a, c, e) show the reconstruction of malaria incidence time series based on the annual cycle (9–15 periodic band) for Nasarawa, Kwara, and Zamfara, respectively.** Right panel (**b, d, f**) display the corresponding phase angles of the annual cycle for each state.

timing of seasonal patterns and in particular, the time when malaria peaks in each of the states. Annual malaria peaks in Nasarawa consistently occurred in September throughout most of the study period, except in 2015 and 2024 when the peak shifted to August, and in 2020 when it occurred in October. In Kwara State, annual malaria peaks occurred mostly in August during the study period, with exceptions in 2015, 2016, and 2024 when peaks occurred earlier, in June, and in 2018 and 2019 when they shifted to September. In Zamfara, annual malaria peaks occurred consistently in September

throughout the entire study period except in 2024 when it peaked in August, indicating an almost stable and predictable seasonal pattern. These reults are presented in Table 1.

## Phase relationships and synchronisation between malaria incidence time series

In this sub-section, we employ the wavelet cross-spectrum and phase difference to compare the association between malaria incidence in Nasarawa and Kwara, Nasarawa and Zamfara, and Zamfara and Kwara States. Fig 5 shows the cross-wavelet power spectrum and the phase difference between these states. The cone of influence, which indicates the region affected by edge effects, is shown with a white line. The colour code for power ranges from blue (low power) to red (high power). The arrows on the cross-wavelet power spectra are helpful in interpreting the phase relationship between any two time series. Arrows pointing to the right mean that the series are in phase. Arrows pointing to the left mean that the series are out of phase. To the right and up, implies that the first series leads and to the left and up, the first series is lagging. To the right and down means the first series lags and to the left and down, the first series is leading [14,58].

For the Nasarawa-Kwara pair, malaria patterns in both states showed similar timing (i.e., they moved together) in the 11–16 month cycle. Between 2015 and mid-2018, Kwara's malaria patterns consistently occurred slightly earlier than those in Nasarawa, indicating a slight but regular lead in timing (Fig 5a). From mid-2018 to early 2020, both states experienced synchronized malaria outbreaks, with peaks and troughs occurring at roughly the same time. From mid-2020–2024, this earlier timing in Kwara re-emerged, with malaria patterns in Kwara again leading those in Nasarawa. However, the features before mid-2016 and after mid-2023 fall within the COI and thus should be treated cautiously. These timing differences are further illustrated in Fig 5b, which captures the lead-lag relationship across time.

For the Nasarawa-Zamfara pair, malaria patterns in both states followed a similar timing during multiple stretches of the study period, particularly in relation to the 6-month transmission cycle: from 2015 to mid-2016, mid-2017, late 2017 to mid-2019, early 2020 to mid-2021, and late 2021 to mid-2023. More importantly, in the stronger 11–14 month cycle (corresponding roughly to annual malaria patterns), the two states remained consistently synchronized throughout the entire study period, howbeit, patterns before mid-2016 and after mid-2023 fall within the COI and should be taken with caution (Fig 5c). The phase difference between malaria incidence in Nasarawa and Zamfara indicates a consistent pattern of synchronization over time (Fig 5d). Throughout the period from 2015 to 2024, the phase difference remains close to 0, confirming that the seasonal peaks and troughs in both states tend to occur almost at the same time. Minor fluctuations around the zero line reflect occasional slight leads or lags, but no state consistently leads or lags the other. A similar malaria pattern observed in the Nasarawa-Kwara pair is also seen in the Zamfara-Kwara pair within the 11–16 month cycle (Fig 5e). This similarity is expected, given that Nasarawa and Zamfara remained in synchrony throughout the study period. Table 2 presents the Pearson correlation coefficients that quantify the synchrony between state-level malaria incidence, based on both the normalized time series and the extracted annual cycles. The correlation based on the normalised series captures all the components and structure of the time series including semi-annual cycles, annual seasonal cycles, multi-annual cycles etc while the correlation based on annual cycle captures only the annual cycle components of each state pair.

**Table 1. Annual peak malaria incidence months (2015–2024) in Nasarawa, Kwara, and Zamfara.**

| State | 2015 | 2016 | 2017 | 2018 | 2019 | 2020 | 2021 | 2022 | 2023 | 2024 |
|---|---|---|---|---|---|---|---|---|---|---|
| Nasarawa | Aug. | Sept. | Sept. | Sept. | Sept. | Oct. | Sept. | Sept. | Sept. | Aug. |
| Kwara | June | June | Aug. | Sept. | Sept. | Aug. | Aug. | Aug. | Aug. | June |
| Zamfara | Sept. | Sept. | Sept. | Sept. | Sept. | Sept. | Sept. | Sept. | Sept. | Aug. |

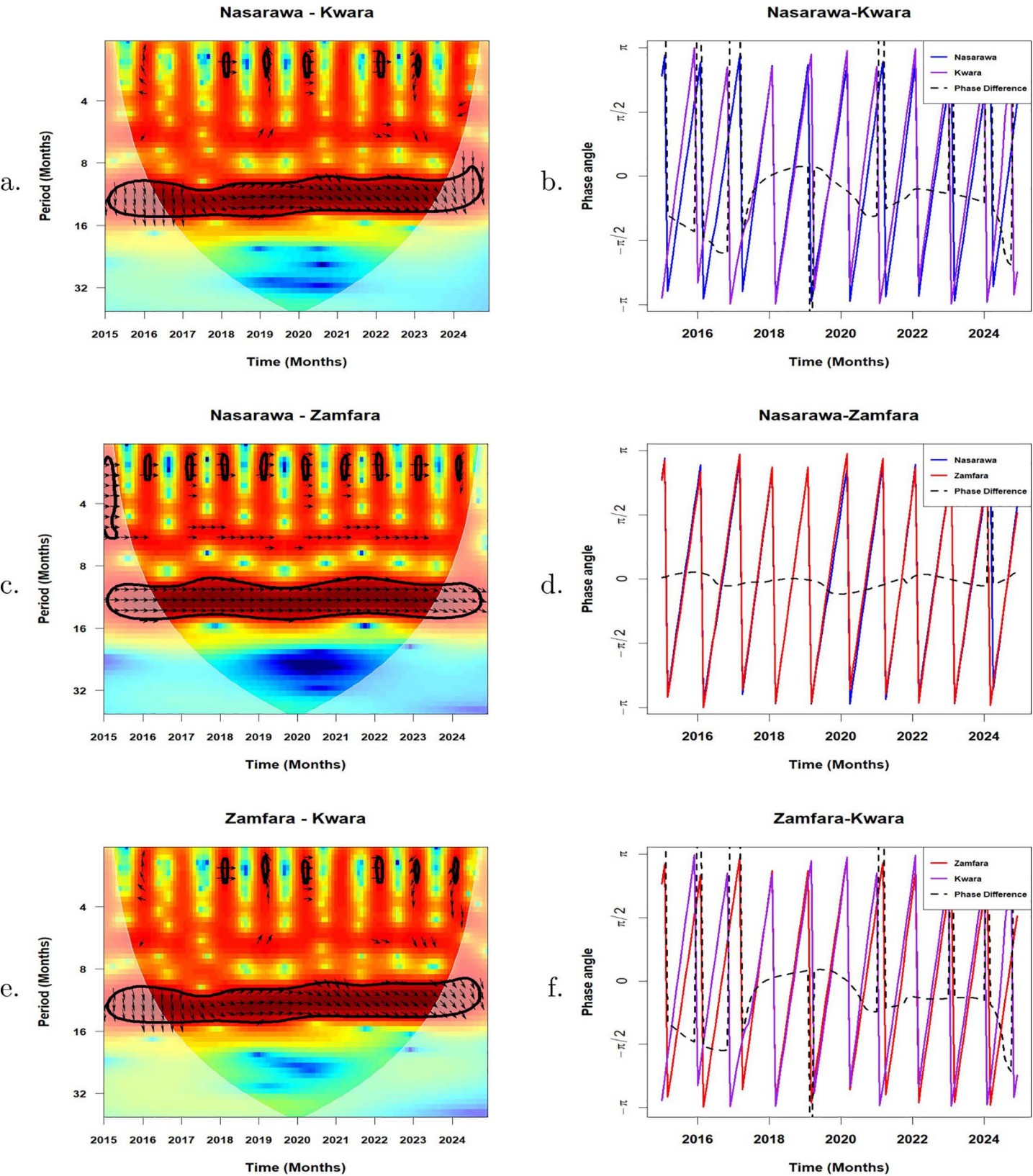

**Fig 5. Left panel: Cross-wavelet power spectrum of malaria incidence between a. Nasarawa and Kwara c. Nasarawa and Zamfara, and e. Zamfara and Kwara.** Right panel: Phase difference between **b.** Nasarawa and Kwara **d.** Nasarawa and Zamfara, and **f.** Zamfara and Kwara. The black

thick contours indicate 5% significance levels against red noise and the white line indicates the cone of influence that delimits the region not influenced by edge effects.

**Table 2. Pearson correlation coefficients quantifying synchrony between state-level malaria incidence, based on both the normalised time series and the annual cycles.**

| State Pair | Distance (km) | Correlation | |
|---|---|---|---|
| | | Normalised series | Annual cycle |
| Nasarawa ⇔ Kwara | 438.31 | 0.5301 | 0.4758 |
| Nasarawa ⇔ Zamfara | 456.31 | 0.5884 | 0.9265 |
| Zamfara ⇔ Kwara | 471.60 | 0.4115 | 0.4169 |

## Forecasting malaria incidence

Logarithmic transformation was applied to the time series in order to reduce heteroscedasticity and improve the stationarity of the data prior to decomposition and modelling. The log-transformed malaria time series for the three states and their corresponding Seasonal-Trend decomposition using Loess (STL) plots are presented in Fig 6. The log-transformed data for each state were split into a training set (80%) spanning January 2015 to December 2022, and a test set (20%) covering January 2023 to December 2024, to facilitate predictive modelling.

The training sets for each state were fitted using the auto.arima() function, which identified optimal candidate models using the AIC. The comparison of these metrics is shown in Fig. 7(a, c, e). The model with the lowest error metrics on this test set was selected as the final model for each state. For Nasarawa State, the SARIMA(0,0,2)(0,1,1) model was selected (MAE = 0.1083, MAPE = 0.0101, RMSE = 0.1546). Similarly, for Kwara State, SARIMA(0,1,2)(1,0,0) demonstrated the lowest error metrics (MAE = 0.1149, MAPE = 0.0114, RMSE = 0.1320). Finally, the SARIMA(1,0,0)(2,1,0) model was chosen for Zamfara, as it yielded the lowest forecast error metrics (MAE = 0.3463, MAPE = 0.0308, RMSE = 0.4026). Fig. 7(b, d, f) displays the resulting forecasts plotted against the observed test data, illustrating the capacity of the final models to track temporal trends.

By using the checkresiduals function in R, diagnostic checks were carried out on the residuals of the three selected models: SARIMA(0,0,2)[0,1,1], SARIMA(0,1,2)[1,0,0] and SARIMA(1,0,0)[2,1,0] for Nasarawa, Kwara and Zamfara States respectively in order to determine whether the estimated models were able to extract adequate information from the data. The Ljung-Box test was used to check for autocorrelation remaining in the residuals. The p-value for the Ljung-Box Q-statistics of the three residuals are 0.45, 0.07 and 0.05 respectively suggesting non-significance. Fig 8(a,c, e) shows the time plots, ACF plots and histograms of the residuals (with an overlaid normal distribution) of the three models. The ACF plot of the residual for SARIMA(0,0,2)[0,1,1] (Fig 8a) shows no autocorrelation suggesting white noise. The ACF plots of the residuals for both SARIMA(0,1,2)[1,0,0] and SARIMA(1,0,0)[2,1,0] (Fig 8c,e) show no significant autocorrelation except for very few lags suggesting minor remaining temporal dependence. The histograms across all three models confirm near-normal residual distributions, centered around zero with mild skewness. These diagnostic results confirm the adequacy of the models.

The selected SARIMA models were used to forecast the malaria cases of each of the three states for 24 months beginning from January 2025 to December 2026. The plots of the forecasted malaria cases are depicted in Fig 8(b, d, f). The forecast line is within a shaded area corresponding to confidence intervals (CI): The 80% confidence interval is represented by the darker coloured area, which shows where the actual values should fall 80% of the time. A larger margin of uncertainty is provided by the 95% confidence interval, which is shown by the lighter grey area. The forecasted log values of malaria cases were back transformed. Tables 3–5 show the forecasted values of malaria cases in Nasarawa, Kwara and Zamfara States respectively from January 2025 to December 2026.

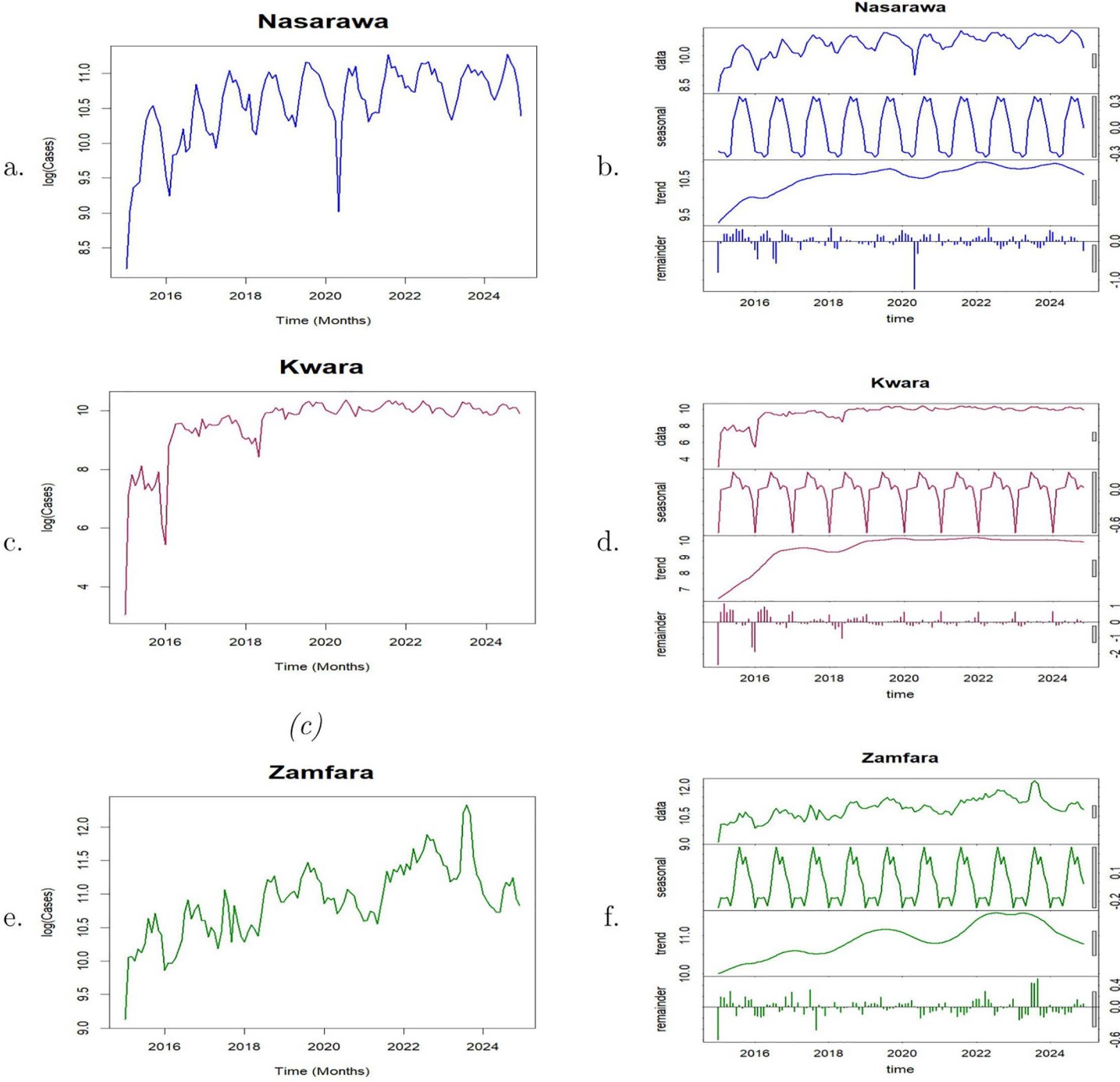

**Fig 6. Left panel: Time series plots of normalised malaria cases in a. Nasarawa, c. Kwara and e. Zamfara States. Right panel: Corresponding decomposition plots for b. Nasarawa d. Kwara and f. Zamfara States.**

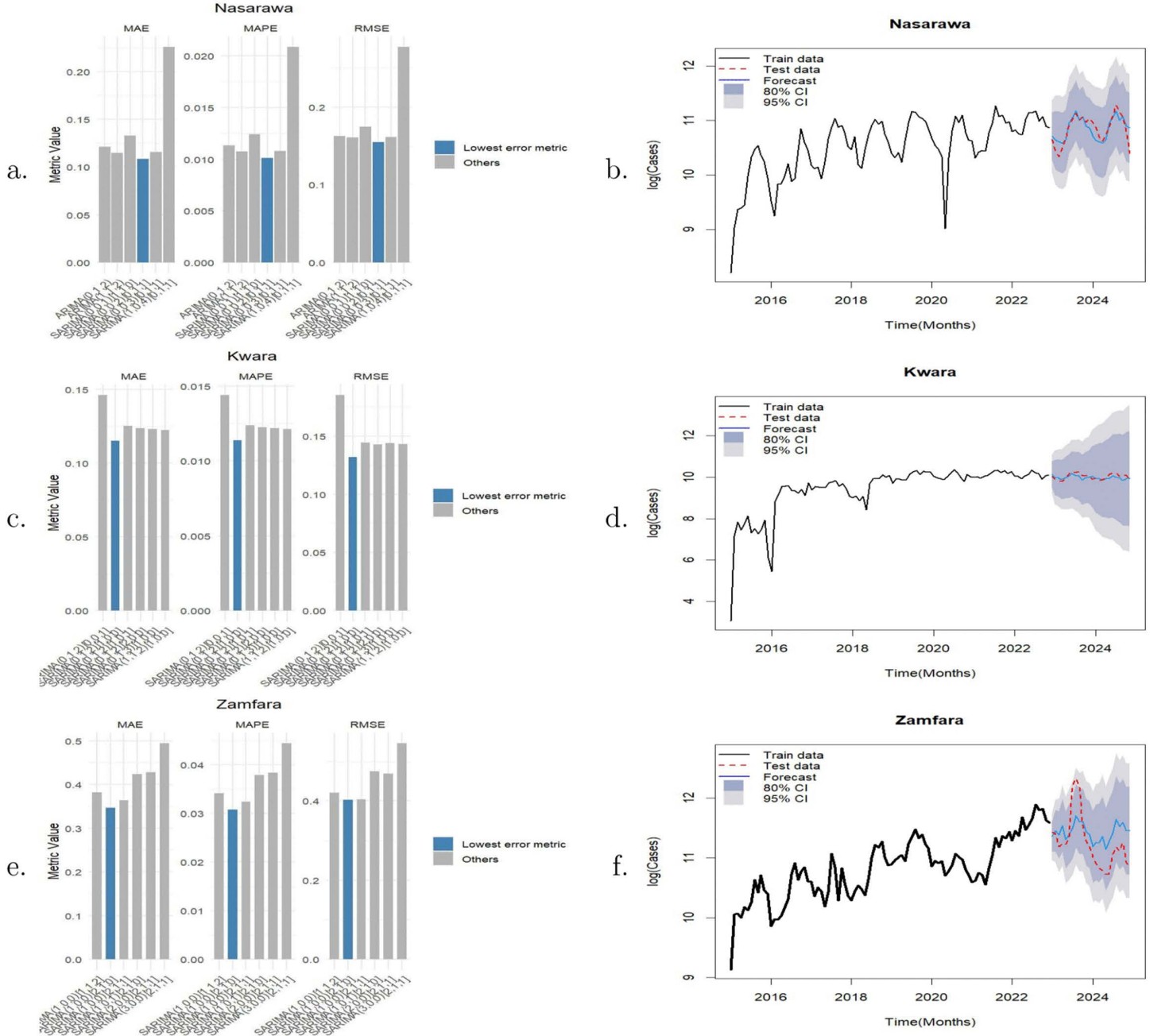

**Fig 7. Evaluation of SARIMA model performance on the 2023–2024 test set.** The left panel (a, c, e) presents a comparison of the forecast error metrics (MAE, MAPE, and RMSE) across candidate models for Nasarawa, Kwara, and Zamfara States, respectively, where the model with the lowest error metrics was selected. The right panel (b, d, f) displays the corresponding time-series forecasts compared against the actual test data.

## Discussions

The wavelet power spectra in Fig 3 reveal multiple periodic signals with important epidemiological implications for the states considered. The strong and persistent 12-month cycle observed in Nasarawa State (Fig 3a) indicates that malaria

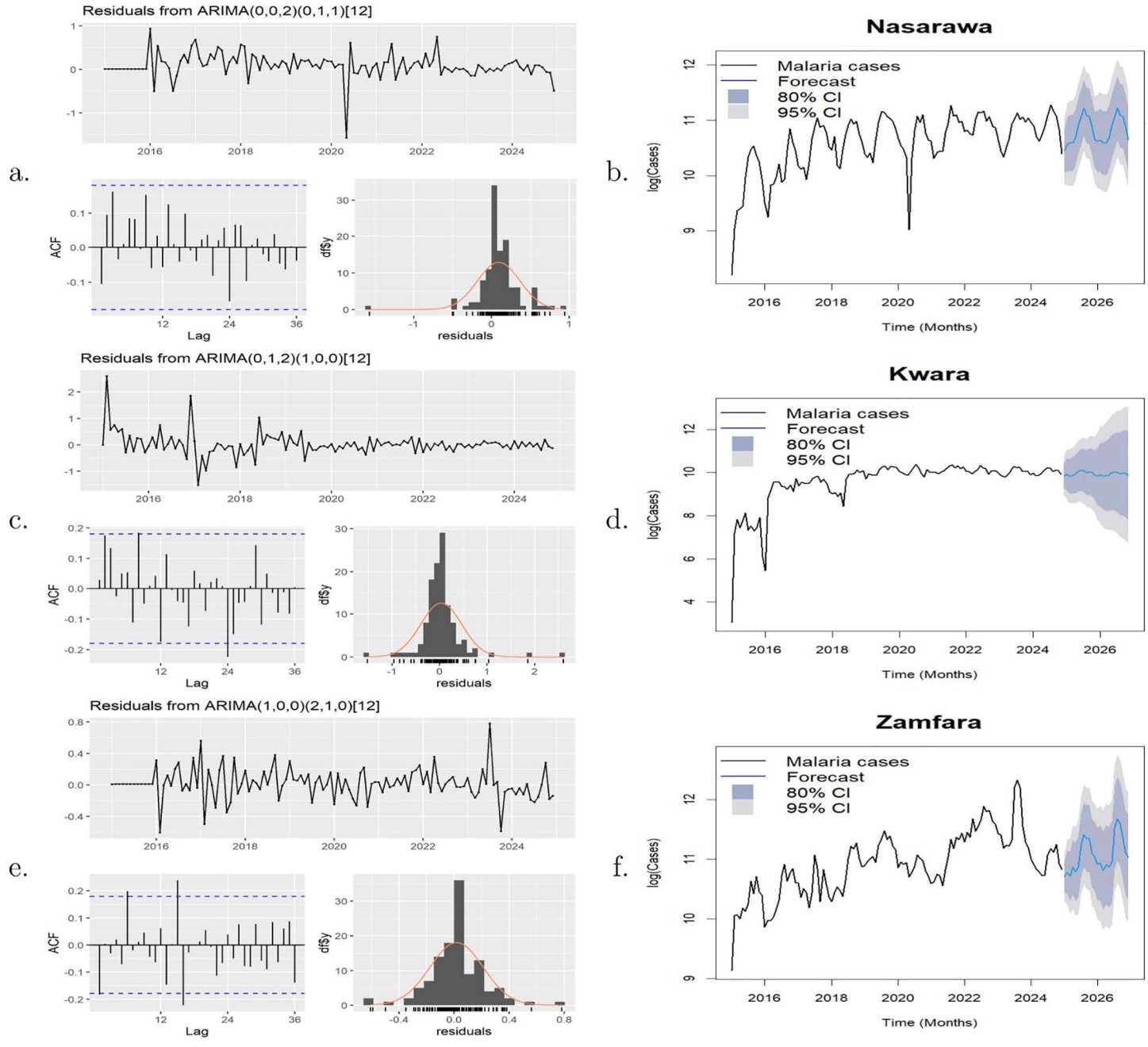

**Fig 8. Left panel: Residuals of Nasarawa, Kwara and Zamfara States.** Right panel: Forecast plots of Nasarawa, Kwara and Zamfara States.

transmission is largely driven by predictable seasonal climatic factors, such as rainfall and temperature. This regular seasonality supports the effective timing of annual interventions, including seasonal malaria chemoprevention (SMC) and long-lasting insecticide nets (LLINs), to coincide with peak transmission, typically in September.

In Kwara State, pronounced semi-annual and annual cycles observed between 2015 and 2017 decline steadily toward the end of the study period, suggesting a reduction in the intensity of seasonal malaria transmission over time. A similar

**Table 3. 24-month forecast of confirmed uncomplicated malaria cases in Nasarawa State.**

| Time | Forecast | Lower 95% CI | Upper 95% CI |
|---|---|---|---|
| Jan 2025 | 34556.18 | 18877.38 | 63257.19 |
| Feb 2025 | 38589.63 | 18318.70 | 81291.74 |
| Mar 2025 | 39764.91 | 18363.05 | 86110.34 |
| Apr 2025 | 39421.70 | 18204.55 | 85367.12 |
| May 2025 | 43951.59 | 20296.41 | 95176.54 |
| Jun 2025 | 54379.06 | 25111.72 | 117757.08 |
| Jul 2025 | 63271.98 | 29218.38 | 137014.57 |
| Aug 2025 | 74432.82 | 34372.34 | 161183.19 |
| Sep 2025 | 65517.08 | 30255.14 | 141876.29 |
| Oct 2025 | 64123.76 | 29611.72 | 138859.08 |
| Nov 2025 | 53106.89 | 24524.24 | 115002.21 |
| Dec 2025 | 41996.22 | 19393.44 | 90942.22 |
| Jan 2026 | 40924.21 | 17636.78 | 94960.11 |
| Feb 2026 | 41573.03 | 17323.03 | 99769.87 |
| Mar 2026 | 39764.91 | 16449.66 | 96126.48 |
| Apr 2026 | 39421.70 | 16307.69 | 95296.81 |
| May 2026 | 43951.59 | 18181.58 | 106247.24 |
| Jun 2026 | 54379.06 | 22495.14 | 131454.28 |
| Jul 2026 | 63271.98 | 26173.90 | 152951.76 |
| Aug 2026 | 74432.82 | 30790.83 | 179931.62 |
| Sep 2026 | 65517.08 | 27102.63 | 158378.98 |
| Oct 2026 | 64123.76 | 26526.26 | 155010.81 |
| Nov 2026 | 53106.89 | 21968.88 | 128378.97 |
| Dec 2026 | 41996.22 | 17372.70 | 101520.39 |

weakening trend is observed for the multi-annual (2.5–3.5-year) cycle. These changes may reflect increasing population immunity, expanded intervention coverage (such as LLINs), or the implementation of SMC. Epidemiologically, this indicates that although malaria transmission in Kwara State previously exhibited strong seasonal and multi-annual patterns, their intensity has waned, pointing to gains in malaria control. Nevertheless, sustained surveillance remains essential to preserve these gains.

For Zamfara State, a strong multi-annual (3–3.5 year) cycle dominates the transmission dynamics, indicating pronounced multi-year variability in malaria incidence. The moderate annual cycle observed between 2015 and late 2019 weakens thereafter, possibly reflecting the scale-up of interventions such as SMC. However, the reappearance and intensification of annual signals from 2022 onward suggest a resurgence of seasonal transmission. This highlights both the effectiveness of sustained interventions and the risk of resurgence if control efforts are not maintained.

Phase relationships and synchronisation between malaria incidence in Nasarawa and Zamfara shows that the two states exhibit potentially very strong synchrony, characterized by a near-zero phase lag. The correlation (based on normalised series) between Nasarawa and Zamfara (0.5884) points to a relatively moderate synchrony in the overall malaria dynamics of the two states. However, for the annual cycles of the two states, a much stronger correlation of 0.9265 is observed confirming the very strong synchrony observed earlier. These results suggest that malaria dynamics in the two states share similar seasonal patterns, timing and structure likely influenced by shared climatic or ecological drivers.

**Table 4. 24-month forecast of confirmed uncomplicated malaria cases in Kwara State.**

| Time | Forecast | Lower 95% CI | Upper 95% CI |
|---|---|---|---|
| Jan 2025 | 18609.32 | 7859.94 | 44059.69 |
| Feb 2025 | 20501.20 | 6481.71 | 64843.84 |
| Mar 2025 | 18677.17 | 5500.48 | 63419.36 |
| Apr 2025 | 19169.34 | 5279.21 | 69605.77 |
| May 2025 | 20271.65 | 5238.00 | 78453.63 |
| Jun 2025 | 23217.38 | 5644.82 | 95494.00 |
| Jul 2025 | 24693.68 | 5663.39 | 107670.22 |
| Aug 2025 | 24160.13 | 5238.53 | 111426.59 |
| Sep 2025 | 21741.55 | 4465.66 | 105851.19 |
| Oct 2025 | 22768.62 | 4438.07 | 116809.81 |
| Nov 2025 | 22387.06 | 4147.85 | 120829.05 |
| Dec 2025 | 19629.99 | 3462.25 | 111296.65 |
| Jan 2026 | 18480.76 | 2448.33 | 139498.58 |
| Feb 2026 | 19828.01 | 2110.77 | 186259.08 |
| Mar 2026 | 18529.71 | 1768.05 | 194196.95 |
| Apr 2026 | 18883.31 | 1622.88 | 219719.83 |
| May 2026 | 19666.42 | 1528.88 | 252974.10 |
| Jun 2026 | 21704.45 | 1532.09 | 307476.77 |
| Jul 2026 | 22698.97 | 1459.83 | 352946.66 |
| Aug 2026 | 22341.47 | 1313.10 | 380124.72 |
| Sep 2026 | 20692.82 | 1114.55 | 384185.11 |
| Oct 2026 | 21398.75 | 1058.92 | 432429.66 |
| Nov 2026 | 21137.54 | 963.23 | 463850.42 |
| Dec 2026 | 19212.02 | 807.95 | 456838.27 |

These findings imply that coordinated preventive interventions such as SMC and LLINs should be deployed around same time across the two states.

Kwara State has a consistent lead over Nasarawa and Zamfara in the timing of malaria incidence. This is possibly due to climatic and geographical variations between Kwara State and the other two states, such as the timing of the rainy season. The correlation, based on the normalised time series, is moderate both for Nasarawa and Kwara (0.5301) and for Zamfara and Kwara (0.4115). Similarly, based on the annual cycles, there is a moderately positive correlation both for Nasarawa and Kwara (0.4758) and for Zamfara and Kwara (0.4169). The lead-lag relationships between these states suggest that malaria dynamics in Kwara State may serve as a precursor for both Nasarawa and Zamfara States. Thus, by observing the commencement and severity of malaria season in Kwara State, public health officials in Nasarawa and Zamfara could gain critical advance notice to prepare for their forthcoming high-transmission season. These results also reveal that synchrony in malaria incidence for the three state pair is not necessarily driven by spatial proximity. Table 2 shows that the Nasarawa–Zamfara pair (456.31 km) have very strong synchrony in the annual cycle compared to Nasarawa–Kwara (438.31 km) although the distance separating Nasarawa and Zamfara is only slightly greater than that between Nasarawa and Kwara. This suggests that shared ecological or climatic factors play a more significant role than distance.

The forecast values for Nasarawa indicate an average of 51,482 (95% CI: 23,899−116,073) monthly malaria cases over the next two years. In 2025, the number of cases is expected to be lowest in January at 34,556 (95% CI:

**Table 5.  24-month forecast of confirmed uncomplicated malaria cases in Zamfara state.**

| Time | Forecast | Lower 95% CI | Upper 95% CI |
|------|----------|--------------|--------------|
| Jan 2025 | 44898.51 | 29922.21 | 67370.58 |
| Feb 2025 | 47982.82 | 27821.01 | 82755.85 |
| Mar 2025 | 45064.68 | 23875.49 | 85059.01 |
| Apr 2025 | 52269.99 | 25972.61 | 105193.59 |
| May 2025 | 48466.86 | 22963.78 | 102293.13 |
| Jun 2025 | 52601.74 | 24037.70 | 115108.46 |
| Jul 2025 | 75100.64 | 33374.62 | 168993.89 |
| Aug 2025 | 90161.20 | 39205.04 | 207346.91 |
| Sep 2025 | 84808.17 | 36252.40 | 198398.58 |
| Oct 2025 | 84190.88 | 35506.05 | 199630.87 |
| Nov 2025 | 66200.44 | 27621.73 | 158661.25 |
| Dec 2025 | 62218.31 | 25740.23 | 150391.74 |
| Jan 2026 | 54758.62 | 21583.27 | 138927.35 |
| Feb 2026 | 55947.45 | 21247.06 | 147319.99 |
| Mar 2026 | 49723.22 | 18345.62 | 134767.82 |
| Apr 2026 | 54456.67 | 19642.41 | 150975.81 |
| May 2026 | 52421.46 | 18574.00 | 147949.30 |
| Jun 2026 | 57145.84 | 19963.97 | 163577.03 |
| Jul 2026 | 100831.66 | 34832.64 | 291882.05 |
| Aug 2026 | 117665.54 | 40286.36 | 343669.19 |
| Sep 2026 | 108228.26 | 36791.92 | 318367.68 |
| Oct 2026 | 85823.49 | 29009.66 | 253904.12 |
| Nov 2026 | 66482.42 | 22369.82 | 197583.75 |
| Dec 2026 | 61657.63 | 20670.73 | 183915.28 |

[a] CI – Confidence Interval.

18,877−63,257) and to peak in August at 74,433 (95% CI: 34,372−161,183). In 2026, the lowest number of cases is expected in April at 39,422 (95% CI: 16,308−95,297), with the peak again occurring in August at 74,433 (95% CI: 30,791−179,932). For Kwara State, the average monthly malaria cases over the same period is estimated at 20,850 (95% CI: 3,381−202,511). In 2025, the lowest forecast is in January at 18,609 (95% CI: 7,860−44,060), rising to a peak of 24,694 (95% CI: 5,663−107,670) in July. In 2026, the forecasted values range from a low of 18,481 (95% CI: 2,448−139,499) in January to a high of 22,699 (95% CI: 1,460−352,947) in July. For Zamfara State, the forecast indicates an average of 67,463 (95% CI: 27,317−171,418) monthly malaria cases over the next two years. The lowest incidence is forecasted for January 2025 (44,899; 95% CI: 29,922–67,371) and March 2026 (49,723; 95% CI: 18,346–134,768), while peak incidence is expected in August 2025 (90,161; 95% CI: 39,205–207,347) and August 2026 (117,666; 95% CI: 40,286–343,669). The 95% CI was used to quantify the forecast uncertainty in order to guide interpretation for the purpose of decision making.

The consistent seasonal peaks from July to August and troughs from January to April indicate a seasonal transmission cycle, probably influenced by rainfall patterns and mosquito breeding patterns. This highlights the fact that malaria dynamics in these states are influenced by climatic and ecological factors. Given the higher forecasted burden in Zamfara State, prioritisation of resources and intensified interventions are warranted, alongside continued preventive efforts in Kwara and Nasarawa States.

The study is however not without its limitations. Firstly, the malaria incidence data used are mostly from public health institutions, and may not reflect the accurate malaria incidence in the states under consideration resulting in underestimation of forecast values. Also, the study period includes major programmatic and contextual changes such as COVID-19 outbreak, and scale-up of SMC and LLINs. However, these potential structural breaks are not explicitly assessed in the study. Another major limitation of the SARIMA model used in the study is its reliance solely on historical data, ignoring other external factors such as intervention coverage rates, policy changes, or climate change that could influence future trends. In order to circumvent this limitation, future studies intends to apply the Seasonal AutoRegressive Integrated Moving Average with Exogenous Variables (SARIMAX) model by incorporating climatic factors and coverage rates of interventions such as SMC and LLINs. Furthermore, the assumption of linearity inherent in SARIMA models restricts their ability to capture nonlinear behaviour that is typical of most epidemiological data.

## Conclusion

In this study, we analyzed 10-year historical malaria data of Nasarawa, Kwara, and Zamfara, three states with different malaria transmission settings in northern Nigeria, to gain useful insights, reveal underlying patterns, and make forecast using wavelet analysis and SARIMA models. Wavelet analysis, suitable for analysing non-stationary data, was used to analyse the transient dynamics of malaria time series data for each of the three states by applying the wavelet power spectrum. The associations between pairs of time series were shown using wavelet cross-spectrum, phase difference, and Pearson correlation.

The wavelet power spectrum shows significant distinctions in malaria seasonality across the three states. While Kwara showed more irregular patterns with alternating semi-annual, annual, and multi-annual cycles with August peaks for nearly half of the study period, Nasarawa's incidence followed a strong annual cycle with peaks occurring in September 70% of the time. With the exception of 2024, when the peak moved to August, Zamfara's yearly cycles were moderate but remarkably stable, peaking in September for the most of the study period. The findings emphasize the need for state-specific malaria control. While the predictable seasonality in Nasarawa and Zamfara allows for accurate timing of interventions, the variability in Kwara calls for more flexibility when deploying interventions.

Malaria incidence in Nasarawa and Zamfara exhibited very strong synchrony, while transmission in Kwara occurred earlier relative to the other two states. Forecasted values for the next 24 months shows an average of approximately 21,000 (95% CI: 3,381−202,511), 51,000 (95% CI: 23,899−116,073) and 67,000 (95% CI: 27,317−171,418) malaria cases in Kwara, Nasarawa and Zamfara respectively. From a public health policy standpoint, very strong synchrony as observed between Nasarawa and Zamfara reflects the need for regionally coordinated control interventions while weaker synchrony such as seen in Kwara and Zamfara necessitates the need for interventions that are locally tailored to individual states. Incorporating synchrony and forecasting analysis into surveillance systems could improve early warning for malaria in all states across Nigeria, optimise resource allocation, and improve the timing of interventions to maximise impact.

This study therefore provides valuable insights into malaria seasonal patterns in the past decade and the future of malaria trends which will in no doubt, help policymakers in making informed decisions on how to effectively plan for ongoing surveillance, timing of resource allocation, and targeted interventions to control malaria spread.

## Acknowledgments

The authors wish to express their sincere appreciation to the anonymous reviewers for their useful suggestions and to Mrs. Temitope Adeyemi, ICAMMDA administrator for helping with typesetting and printing some materials in the course of carrying out this project.

## Author contributions

**Conceptualization:** Emmanuel Afolabi Bakare.

**Data curation:** Emmanuel Afolabi Bakare, Oluwaseun Akinlo Mogbojuri, Dolapo Ayomide Bakare, Oluwakemi Janet Odewusi.

**Formal analysis:** Emmanuel Afolabi Bakare, Oluwaseun Akinlo Mogbojuri, Dolapo Ayomide Bakare, Oluwakemi Janet Odewusi.

**Funding acquisition:** Emmanuel Afolabi Bakare.

**Investigation:** Emmanuel Afolabi Bakare, Oluwakemi Janet Odewusi.

**Methodology:** Emmanuel Afolabi Bakare, Dolapo Ayomide Bakare.

**Project administration:** Emmanuel Afolabi Bakare, Oluwaseun Akinlo Mogbojuri.

**Resources:** Emmanuel Afolabi Bakare, Oluwaseun Akinlo Mogbojuri.

**Software:** Emmanuel Afolabi Bakare, Dolapo Ayomide Bakare.

**Supervision:** Emmanuel Afolabi Bakare.

**Validation:** Emmanuel Afolabi Bakare.

**Visualization:** Emmanuel Afolabi Bakare, Oluwaseun Akinlo Mogbojuri, Dolapo Ayomide Bakare, Oluwakemi Janet Odewusi.

**Writing – original draft:** Emmanuel Afolabi Bakare, Oluwaseun Akinlo Mogbojuri, Dolapo Ayomide Bakare, Oluwakemi Janet Odewusi.

**Writing – review & editing:** Emmanuel Afolabi Bakare.

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
