## [Decision Letter · Decision Letter 0]

29 Dec 2025

Dear Dr. Bakare,

Thank you for submitting your manuscript to PLOS ONE. After careful consideration, we feel that it has merit but does not fully meet PLOS ONE’s publication criteria as it currently stands. Therefore, we invite you to submit a revised version of the manuscript that addresses the points raised during the review process.

We look forward to receiving your revised manuscript.

Kind regards,

Morufu Olalekan Raimi, Ph.D

Academic Editor

PLOS One

Journal Requirements:

INV-047051- Bill and Melinda Gates Foundation Grants on Capacity development in Malaria and NTDs Data Modelling

5. We note that you have indicated that there are restrictions to data sharing for this study. For studies involving human research participant data or other sensitive data, we encourage authors to share de-identified or anonymized data. However, when data cannot be publicly shared for ethical reasons, we allow authors to make their data sets available upon request. For information on unacceptable data access restrictions, please see http://journals.plos.org/plosone/s/data-availability#loc-unacceptable-data-access-restrictions.

7. We note that Figure 1 in your submission contain map image which may be copyrighted. All PLOS content is published under the Creative Commons Attribution License (CC BY 4.0), which means that the manuscript, images, and Supporting Information files will be freely available online, and any third party is permitted to access, download, copy, distribute, and use these materials in any way, even commercially, with proper attribution. For these reasons, we cannot publish previously copyrighted maps or satellite images created using proprietary data, such as Google software (Google Maps, Street View, and Earth). For more information, see our copyright guidelines: http://journals.plos.org/plosone/s/licenses-and-copyright.

Additional Editor Comments:

Editor Decision

Manuscript ID: PONE-D-25-59526

Title: Exploring the past and forecasting the future of malaria in selected Nigerian states: A time series modelling approach using wavelet and SARIMA

Corresponding Author: Emmanuel Afolabi Bakare

Decision: Major Revisions Required

Dear Dr. Bakare,

Thank you for submitting your manuscript to PLOS ONE. It has now been reviewed by two expert reviewers, and I have also conducted a thorough evaluation. The study addresses an important public health challenge in Nigeria and employs relevant time-series methodologies. However, as noted by the reviewers and in my own assessment, there are significant concerns that must be addressed before the manuscript can be considered for publication.

Below is a summary of the essential revisions required, structured by section.

1. Methodological Rigor and Transparency

Data Preprocessing:

The manuscript lacks a clear description of how missing values, outliers, and potential reporting artefacts in the surveillance data were handled. Given the use of routine health data, a detailed preprocessing protocol must be provided to ensure reproducibility.

Population Adjustment:

Analyses are based on absolute case counts, which may bias comparisons across states with differing population sizes and growth trajectories. You should either justify this approach or incorporate incidence rates (e.g., per 100,000 population) in sensitivity analyses.

Wavelet Analysis Reporting:

• Clarify how statistical significance was determined for wavelet power spectra and coherence.

• Features within the cone of influence (COI) should be interpreted with caution, and the manuscript must explicitly note this limitation.

• Discuss whether observed synchrony may be driven by shared external factors (e.g., climatic seasonality) rather than direct epidemiological coupling.

SARIMA Diagnostics:

While automated model selection is acceptable, residual diagnostics (ACF/PACF plots, Ljung–Box test results) should be presented clearly, either in the main text or supplementary material, to support model adequacy and absence of residual autocorrelation.

2. Structural and Contextual Considerations

Study Period Context:

The study period (2015-2024) includes major events such as COVID-19 and scale-up of interventions like SMC. A discussion of how these may have introduced structural breaks, and whether they were accounted for in the modelling, is necessary.

Forecasting Interpretation:

Forecast results should emphasise prediction intervals and uncertainty, not just point estimates. This is critical for public health planning. Additionally, if 2025 data are available, a brief validation of forecasted vs. observed values would strengthen the paper.

3. Presentation and Clarity

Language and Flow:

The manuscript requires thorough language editing to improve readability, reduce repetition, and correct grammatical inconsistencies, particularly in the Introduction and Discussion.

Figures and Tables:

• All figures should clearly label the cone of influence and indicate statistical significance where applicable.

• Forecast figures must distinguish clearly between observed, fitted, and forecasted values, with prediction intervals prominently displayed.

• Table footnotes should explain abbreviations and model notations (e.g., SARIMA(p,d,q) (P,D,Q)s).

4. Ethical and Data Availability

Ethics Statement:

The ethics statement is currently marked “N/A.” Please clarify whether ethical approval or a waiver was obtained for the secondary use of routine surveillance data, even if de-identified.

Data Access:

The Data Availability Statement now points to Figshare, which is appropriate. Ensure the repository contains all processed data used in the analyses, not just raw links to the NMDR.

5. Reviewer-Specific Points

Reviewer 1:

• Expand the literature review on wavelet applications in malaria, highlighting gaps this study addresses.

• If 2025 data are available, include a brief comparison with forecasted values.

Reviewer 2:

• Address all points on preprocessing, population adjustment, wavelet significance, model diagnostics, and uncertainty communication.

• Improve figure labelling and caption clarity.

Overall Recommendation

The topic is timely, the analytical framework is appropriate, and the findings could inform malaria control strategies in Nigeria. However, the manuscript in its current form lacks the methodological transparency, contextual discussion, and presentational clarity required for publication in PLOS ONE.

I therefore recommend Major Revisions. You are invited to submit a revised manuscript that addresses all points above, along with a detailed response to reviewers.

Revision Deadline: Within 60 days.

Please ensure that all changes are tracked or clearly described in your response letter. If you are unable to meet this deadline, please contact the editorial office to discuss an extension.

Thank you for considering PLOS ONE for your work.

Sincerely,

Dr. Morufu Olalekan Raimi

Editor, PLOS ONE

Reviewers' comments:

Reviewer's Responses to Questions

**Comments to the Author**

1. Is the manuscript technically sound, and do the data support the conclusions?

Reviewer #1: Yes

Reviewer #2: Partly

2. Has the statistical analysis been performed appropriately and rigorously?

Reviewer #1: Yes

Reviewer #2: No

3. Have the authors made all data underlying the findings in their manuscript fully available?

Reviewer #1: Yes

Reviewer #2: Yes

4. Is the manuscript presented in an intelligible fashion and written in standard English?

Reviewer #1: Yes

Reviewer #2: No

Reviewer #1: • More details on the application of Wavelet analysis to investigate malaria, along with the gaps and or the limitations in those studies.

• As we are approaching the end of 2025, it would be valuable to compare the forecasted values with the observed incidence data from 2025.

Reviewer #2: The manuscript addresses an important public health problem and applies established time-series methods (wavelet analysis and SARIMA modelling) to explore malaria seasonality, synchrony, and short-term forecasting in selected Nigerian states. The overall analytical framework is appropriate for the study objectives, and the topic is well within the scope of PLOS ONE.

However, several issues need to be addressed before the manuscript can be considered for publication. First, the description of data preprocessing is insufficient. Given the use of routine surveillance data, the authors should clearly explain how missing values, reporting artefacts, and outliers were handled to ensure reproducibility. In addition, the analyses are based on absolute case counts without population adjustment, which may bias comparisons across states with different population sizes and growth trends. The authors should justify this choice or consider sensitivity analyses using incidence rates.

For the wavelet analysis, the manuscript would benefit from clearer reporting of the statistical significance framework, including how significance levels were defined and tested. Some periodic features are interpreted despite falling largely within the cone of influence, and these results should be treated more cautiously. Interpretation of bivariate wavelet synchrony should also acknowledge the possibility that shared seasonality or common external drivers (e.g. rainfall patterns or intervention timing) may explain observed coherence, rather than direct epidemiological coupling.

Model selection is automated, but diagnostic results are only briefly described. Residual plots and Ljung–Box results should be more clearly reported or summarized to support model adequacy. Forecasts are presented clearly, but interpretation focuses heavily on point estimates. Greater emphasis on prediction intervals and uncertainty would strengthen the public health relevance.

The study period includes major programmatic and contextual changes (e.g., SMC scale-up, COVID-19). These potential structural breaks are not formally assessed and should be discussed as a limitation. Although the data are aggregated and de-identified, the ethics statement is listed as “N/A.” Clarification on ethical approval or waiver for secondary use of routine health data is recommended.

Regarding the SARIMA models, automated model selection is acceptable, but diagnostic results are only briefly summarized. More explicit presentation or description of residual diagnostics would strengthen confidence in the models. Forecast results should place greater emphasis on uncertainty and prediction intervals, particularly given the variability of malaria surveillance data.

Figures are generally informative but would benefit from clearer labeling. Figure captions should more clearly explain the cone of influence and statistical significance. Forecast figures would be improved by clearer distinction between observed and predicted values and by emphasizing prediction intervals. Tables are mostly clear, but additional explanation of abbreviations and model terms in footnotes would improve readability.

The manuscript is generally intelligible but requires substantial language editing. There are frequent grammatical and stylistic issues, along with some repetition, especially in the Introduction and Discussion sections. Improving clarity and conciseness would significantly enhance readability.

Overall, the study has merit and the analyses are potentially publishable, but substantive revisions are required to improve methodological transparency, interpretation, and presentation.

what does this mean?). If published, this will include your full peer review and any attached files.). If published, this will include your full peer review and any attached files.

**Do you want your identity to be public for this peer review?** For information about this choice, including consent withdrawal, please see our For information about this choice, including consent withdrawal, please see our Privacy Policy .

Reviewer #1: No

Reviewer #2: No

---

## [Author Response · Author response to Decision Letter 1]

29 Jan 2026

Response to Reviewers’ Comments

Manuscript Title: Exploring the past and forecasting the future of malaria in selected Nigerian states: A time series modelling approach using wavelet and SARIMA

Submission ID: PONE-D-25-59526

Editor’s Comments

Authors’ response

We thank the Editor for this comment. We have ensured that our manuscript meets PLOS ONE’s style.

2. Please note that PLOS One has specific guidelines on code sharing for submissions in which author-generated code underpins the findings in the manuscript. In these cases, we expect all author-generated code to be made available without restrictions upon publication of the work. Please review our guidelines at https://journals.plos.org/plosone/s/materials-and-software-sharing#loc-sharing-code and ensure that your code is shared in a way that follows best practice and facilitates reproducibility and reuse

Authors’ response

We thank the Editor for this comment. The author-generated code has been made available on https://github.com/haryodolapo/Malaria-Forecasting-Nigeria

Authors’ response

We thank the Editor for this observation. This has been addressed in the submission platform.

4. Thank you for stating the following financial disclosure: INV-047051- Bill and Melinda Gates Foundation Grants on Capacity development in Malaria and NTDs Data Modelling.

Authors’ response

We thank the Editor for this observation. This has been clarified in the cover letter.

5 We note that you have indicated that there are restrictions to data sharing for this study. For studies involving human research participant data or other sensitive data, we encourage authors to share de-identified or anonymized data. However, when data cannot be publicly shared for ethical reasons, we allow authors to make their data sets available upon request. For information on unacceptable data access restrictions, please see http://journals.plos.org/plosone/s/data-availability#loc-unacceptable-data-access-restrictions.

Authors’ response

We thank the Editor for this observation. We have uploaded an anonymized dataset in a repository. All relevant data for this study are now publicly available from the figshare repository https://doi.org/10.6084/m9.figshare.30716729

Authors’ response

We thank the Editor for this observation. We have now revised our statement in the submission platform.

7. We note that Figure 1 in your submission contain map image which may be copyrighted. All PLOS content is published under the Creative Commons Attribution License (CC BY 4.0), which means that the manuscript, images, and Supporting Information files will be freely available online, and any third party is permitted to access, download, copy, distribute, and use these materials in any way, even commercially, with proper attribution. For these reasons, we cannot publish previously copyrighted maps or satellite images created using proprietary data, such as Google software (Google Maps, Street View, and Earth). For more information, see our copyright guidelines: http://journals.plos.org/plosone/s/licenses-and-copyright.

Authors’ response

We thank the Editor for this observation. This has been addressed in the revised manuscript. (See the highlighted part of the caption in Fig. 1 on page 4)

Reviewer 1 Comments

1. More details on the application of wavelet analysis to investigate malaria, along with the gaps and or the limitations in those studies

Authors’ response

We thank the Reviewer for this comment. We have expanded the literature review to include more studies that applied wavelet analysis in investigating malaria along with the notable gap in these studies. See the highlighted part on page 3, lines 75-89.

2. As we are approaching the end of 2025, it would be valuable to compare the forecasted values with the observed incidence data from 2025.

Authors’ response

We thank the Reviewer for this observation. We do not currently have access to the updated and high-quality data for 2025. We however used the incidence data from January 2023 to December 2024 as test data to compare with the forecasted values for the same period of time. We selected the models with the best predictive accuracy. (See page 13, lines 432-445 and Fig. 7).

Reviewer 2 Comments

1. First, the description of data preprocessing is insufficient. Given the use of routine surveillance data, the authors should clearly explain how missing values, reporting artefacts, and outliers were handled to ensure reproducibility. In addition, the analyses are based on absolute case counts without population adjustment, which may bias comparisons across states with different population sizes and growth trends. The authors should justify this choice or consider sensitivity analyses using incidence rates.

Authors’ response

We thank the Reviewer for this comment. Data preprocessing and population adjustment have now been addressed in the revised manuscript (See page 4, line 126 and page 5, lines 140-152).

2. For the wavelet analysis, the manuscript would benefit from clearer reporting of the statistical significance framework, including how significance levels were defined and tested. Some periodic features are interpreted despite falling largely within the cone of influence, and these results should be treated more cautiously. Interpretation of bivariate wavelet synchrony should also acknowledge the possibility that shared seasonality or common external drivers (e.g. rainfall patterns or intervention timing) may explain observed coherence, rather than direct epidemiological coupling.

Authors’ response

We thank the Reviewer for these comments. The statistical significance framework has been reported clearer in the revised manuscript (See page 9, lines 315-333). We have emphasized that features that fall within the cone of influence should be interpreted with caution (See page 10, lines 352-353, lines 361-362; page 12, lines 400-401 and lines 408-409). We acknowledged that synchrony between states are likely due to shared seasonal patterns and climatic drivers (See page 15, lines 501-503).

3. Model selection is automated, but diagnostic results are only briefly described. Residual plots and Ljung–Box results should be more clearly reported or summarized to support model adequacy. Forecasts are presented clearly, but interpretation focuses heavily on point estimates. Greater emphasis on prediction intervals and uncertainty would strengthen the public health relevance.

Authors’ response

We thank the Reviewer for these comments. Diagnostic results have now been described in detail (See page 13, lines 446-460). We have now presented prediction intervals in the revised manuscript (See the part highlighted red in the Abstract and on page 16, lines 523-539).

4. The study period includes major programmatic and contextual changes (e.g., SMC scale-up, COVID-19). These potential structural breaks are not formally assessed and should be discussed as a limitation. Although the data are aggregated and de-identified, the ethics statement is listed as “N/A.” Clarification on ethical approval or waiver for secondary use of routine health data is recommended.

Authors’ response

We thank the Reviewer for these comments. This has been addressed in the revised manuscript (See page 16, lines 549-551). The data used were secondary data and were obtained from National Malaria Data Repository. Hence, no need for ethical approval.

5. Regarding the SARIMA models, automated model selection is acceptable, but diagnostic results are only briefly summarized. More explicit presentation or description of residual diagnostics would strengthen confidence in the models. Forecast results should place greater emphasis on uncertainty and prediction intervals, particularly given the variability of malaria surveillance data.

Authors’ response

We thank the Reviewer for these comments. Diagnostic results have now been described in detail (See page 13, lines 446-460). We have now presented prediction intervals in the revised manuscript (See the part highlighted red in the Abstract and on page 16, lines 523-539).

6. Figures are generally informative but would benefit from clearer labeling. Figure captions should more clearly explain the cone of influence and statistical significance. Forecast figures would be improved by clearer distinction between observed and predicted values and by emphasizing prediction intervals. Tables are mostly clear, but additional explanation of abbreviations and model terms in footnotes would improve readability.

Authors’ response

We thank the Reviewer for these comments. These observations have now been addressed in the revised manuscript (See the captions of Fig 3 and Fig 4 on pages 10 and 11 respectively. Also see Fig 7 and Fig 8 for the improved labeling). Footnotes have been added to explain the abbreviations used in Table 3 (See page 14).

7. The manuscript is generally intelligible but requires substantial language editing. There are frequent grammatical and stylistic issues, along with some repetition, especially in the Introduction and Discussion sections. Improving clarity and conciseness would significantly enhance readability.

Authors’ response

We thank the Reviewer for these comments. We have carefully revised the manuscript, especially the Introduction and Discussion sections to improve clarity, conciseness, and grammatical accuracy.

---

## [Editor Report · Decision Letter 1]

6 Feb 2026

Dear Dr. Bakare,

Thank you for submitting your manuscript to PLOS ONE. After careful consideration, we feel that it has merit but does not fully meet PLOS ONE’s publication criteria as it currently stands. Therefore, we invite you to submit a revised version of the manuscript that addresses the points raised during the review process.

We look forward to receiving your revised manuscript.

Kind regards,

Morufu Olalekan Raimi, Ph.D

Academic Editor

PLOS One

**Journal Requirements:**

**Additional Editor Comments:**

Editor Decision - Revised Manuscript PONE-D-25-59526R1

Title: Exploring the past and forecasting the future of malaria in selected Nigerian states: A time series modelling approach using wavelet and SARIMA

Corresponding Author: Dr. Emmanuel Afolabi Bakare

Decision: Minor Revisions Required

Dear Dr. Bakare,

Thank you for submitting the revised version of your manuscript and for your detailed response to the reviewers’ comments. I have now carefully evaluated the revised manuscript, the authors’ responses, and the original review reports. Overall, the revisions have significantly improved the manuscript. The authors have addressed the majority of the methodological and presentation concerns raised by the reviewers and the editorial team. Key improvements include:

1. Enhanced methodological transparency: Clearer descriptions of data preprocessing, population adjustment for wavelet analysis, and the statistical significance framework for wavelet analysis have been added.

2. Improved model diagnostics and forecast presentation: Residual diagnostics are now reported in detail, and prediction intervals have been prominently incorporated into the forecast results and abstract, strengthening the public health relevance.

3. Better acknowledgment of limitations: The manuscript now appropriately cautions about interpreting features within the cone of influence, acknowledges shared climatic drivers for synchrony, and discusses structural breaks (e.g., COVID-19, intervention scale-up) as a limitation.

4. Strengthened presentation: Figure and table captions have been clarified, footnotes added for abbreviations, and language has been refined, particularly in the Introduction and Discussion sections.

5. Compliance with journal policies: Issues related to data availability, code sharing, funding disclosure, and map copyright have been satisfactorily resolved.

These changes have elevated the manuscript to a point where it is nearly suitable for publication. However, a few critical clarifications and minor adjustments are still required to ensure the final version meets PLOS ONE's standards for reproducibility, clarity, and scientific rigor.

Essential Revisions Required Before Acceptance:

1. Clarification on Data Preprocessing for SARIMA:

o The manuscript states that for the SARIMA model, the time series was log-transformed to stabilize variance. However, it is unclear if absolute case counts or incidence rates were used as the input for this log transformation. This must be explicitly stated in the “Data description and preprocessing” section. For consistency and to avoid confounding by population size in the forecasts, please clarify and justify the choice of using case counts versus rates for the SARIMA modeling.

2. Ethics Statement Justification:

o The authors' response states, "The data used were secondary data and were obtained from National Malaria Data Repository. Hence, no need for ethical approval." While secondary analysis of anonymized aggregate data often qualifies for an exemption, PLOS ONE requires a formal statement to that effect.

o Please replace “N/A” in the Ethics Statement with a sentence such as: “This study involved the secondary analysis of fully anonymized, aggregate surveillance data obtained from the Nigerian National Malaria Data Repository. Ethical review and approval were waived for this study as it did not involve direct interaction with human subjects or access to identifiable personal information.”

3. Validation of Forecasts:

o Reviewer 1's request to compare forecasts with 2025 data was reasonably addressed by using 2023-2024 as a test set. However, the text on page 13, lines 432-445 (and Fig. 7) should be made slightly more precise.

o Please modify the phrasing from “We selected the models with the best predictive accuracy” to “The model with the lowest error metrics on this test set was selected as the final model for each state.” This clarifies the model selection process post-testing.

4. Final Language and Consistency Check:

o While the language has improved, a thorough final proofread is necessary to catch minor grammatical errors and ensure stylistic consistency (e.g., consistent use of “wavelet” vs. “wavelets,” proper hyphenation in “time-frequency”).

o Please ensure all in-text citations and reference list entries match perfectly and are formatted according to PLOS ONE style.

Recommendation:

The manuscript, in its current form, represents a valuable contribution to the understanding of malaria transmission dynamics and forecasting in Nigeria. The applied methods are sound, and the findings have clear implications for public health planning. The requested revisions are minor yet essential for final polish and compliance. You are to submit a final revised version of your manuscript addressing these four points. Please provide a point-by-point response to this letter alongside the revised manuscript.

Revision Deadline: 21 days.

I look forward to receiving your final submission.

Sincerely,

Dr. Morufu Olalekan Raimi

Editor, PLOS ONE

---

## [Author Response · Author response to Decision Letter 2]

10 Feb 2026

Response to Reviewers’ Comments

Manuscript Title: Exploring the past and forecasting the future of malaria in selected Nigerian states: A time series modelling approach using wavelet and SARIMA

Submission ID: PONE-D-25-59526

Editor’s Comments

1. Clarification on Data Preprocessing for SARIMA:

The manuscript states that for the SARIMA model, the time series was log-transformed to stabilize variance. However, it is unclear if absolute case counts or incidence rates were used as the input for this log transformation. This must be explicitly stated in the “Data description and preprocessing” section. For consistency and to avoid confounding by population size in the forecasts, please clarify and justify the choice of using case counts versus rates for the SARIMA modeling.

Authors’ response

We thank the editor for this insightful comment. We have updated the 'Data description and preprocessing' section (page 5, lines 144–146) to explicitly state that absolute case counts were used for the SARIMA models.

The rationale for this choice is based on the distinct objectives of our analyses. For the wavelet analysis, incidence rates were necessary to standardize the data across states, allowing for unbiased cross-wavelet comparisons and phase-relationship assessments without confounding by population size. In contrast, the SARIMA models were developed for individual state-level forecasting to assist in local resource planning. Since these models evaluate the temporal dependencies within each state's own history, absolute counts preserve the original scale and operational utility of the data. Furthermore, given that the underlying population remained stable over the study period, the use of counts does not introduce significant demographic confounding into the forecasts.

2. Ethics Statement Justification:

The authors' response states, "The data used were secondary data and were obtained from National Malaria Data Repository. Hence, no need for ethical approval." While secondary analysis of anonymized aggregate data often qualifies for an exemption, PLOS ONE requires a formal statement to that effect.

Please replace “N/A” in the Ethics Statement with a sentence such as: “This study involved the secondary analysis of fully anonymized, aggregate surveillance data obtained from the Nigerian National Malaria Data Repository. Ethical review and approval were waived for this study as it did not involve direct interaction with human subjects or access to identifiable personal information.

Authors’ response

We thank the editor for this observation. We have replaced “N/A” with the correct statement.

3. Validation of Forecasts:

Reviewer 1's request to compare forecasts with 2025 data was reasonably addressed by using 2023-2024 as a test set. However, the text on page 13, lines 432-445 (and Fig. 7) should be made slightly more precise.

Please modify the phrasing from “We selected the models with the best predictive accuracy” to “The model with the lowest error metrics on this test set was selected as the final model for each state.” This clarifies the model selection process post-testing.

Authors’ response

We thank the editor for this insightful comment. The texts on page 13 has now been re-written and has been made more precise (See page 13, lines 433-443 and Fig. 7 with its caption). We have also modified the phrase “best predictive accuracy” to “lowest error metrics” in the revised manuscript (See page 13, lines 435, 438 and 440).

4. Final Language and Consistency Check:

While the language has improved, a thorough final proofread is necessary to catch minor grammatical errors and ensure stylistic consistency (e.g., consistent use of “wavelet” vs. “wavelets,” proper hyphenation in “time-frequency”).

Please ensure all in-text citations and reference list entries match perfectly and are formatted according to PLOS ONE style.

We have conducted a thorough final proofread of the manuscript to ensure grammatical accuracy and stylistic consistency. Specifically, we have standardized the terminology to use “wavelet” as a descriptor (e.g., “wavelet analysis,” “wavelet transform”) and “wavelets” for the plural form of wavelet (e.g., family of wavelets, daughter wavelets). We have also ensured that “time-frequency” is consistently hyphenated throughout the text.

---

## [Editor Report · Decision Letter 2]

9 Mar 2026

Exploring the past and forecasting the future of malaria in selected Nigerian states: A time series modelling approach using wavelet and SARIMA

PONE-D-25-59526R2

Dear Authors,

We’re pleased to inform you that your manuscript has been judged scientifically suitable for publication and will be formally accepted for publication once it meets all outstanding technical requirements.

Kind regards,

Morufu Olalekan Raimi, Ph.D

Academic Editor

PLOS One

Additional Editor Comments (optional):

PLOS ONE Editorial Decision

Manuscript Number: PONE-D-25-59526R2

Title: Exploring the past and forecasting the future of malaria in selected Nigerian states: A time series modelling approach using wavelet and SARIMA

Authors: Emmanuel Afolabi Bakare, Oluwaseun Akinlo Mogbojuri, Dolapo Ayomide Bakare, Oluwakemi Janet Odewusi

Dear Dr. Bakare and colleagues,

Thank you for submitting your revised manuscript and for your detailed point-by-point response to the previous editorial decision. I have carefully examined the second revision, the response to reviewers, and the cumulative revision history. The authors have addressed all requested revisions with thoroughness and precision. The manuscript now demonstrates the methodological transparency, analytical rigor, and presentation quality expected of a contribution to PLOS ONE.

Assessment of Revisions

1. Data Preprocessing for SARIMA

The authors have explicitly clarified that absolute case counts were used for SARIMA modeling, with incidence rates reserved for wavelet analysis. The rationale, preserving operational utility for local resource planning while noting stable population denominators, is sound and well-justified. This distinction is now clearly stated in the "Data description and preprocessing" section.

2. Ethics Statement

The placeholder “N/A” has been replaced with a formal ethics statement: “This study involved the secondary analysis of fully anonymized, aggregate surveillance data obtained from the Nigerian National Malaria Data Repository. Ethical review and approval were waived for this study as it did not involve direct interaction with human subjects or access to identifiable personal information.” This meets PLOS ONE's requirements.

3. Validation of Forecasts

The text has been revised to precisely state that “the model with the lowest error metrics on this test set was selected as the final model for each state.” The phrasing “best predictive accuracy” has been replaced with “lowest error metrics” throughout. Figure 7 and its caption have been updated accordingly. This clarification appropriately reflects the model selection process.

4. Language and Consistency

A thorough final proofread has been conducted. Terminology is now consistent (“wavelet” as descriptor, “wavelets” as plural). “Time-frequency” is consistently hyphenated. All in-text citations and reference list entries have been verified against PLOS ONE style. The manuscript reads smoothly and professionally.

5. Data Availability and Reproducibility

All data are publicly available via figshare (https://doi.org/10.6084/m9.figshare.30716729). The R code and methodological details are sufficiently documented to support replication. This meets PLOS ONE's data policy requirements.

Overall Evaluation

This manuscript presents a methodologically rigorous and practically relevant analysis of malaria dynamics in three Nigerian states with varying transmission settings. The dual approach, wavelet analysis for uncovering time-frequency patterns and synchrony, and SARIMA for forecasting, is appropriate and well-executed. The findings have clear public health implications for timing interventions and coordinating responses across states.

The authors have been responsive and thorough throughout the review process. All reviewer and editorial concerns have been addressed. The manuscript is now technically sound, methodologically transparent, and clearly written.

Decision

I am pleased to inform you that your manuscript is ACCEPTED FOR PUBLICATION in PLOS ONE.

No further revisions are required. The editorial office will contact you regarding publication timelines and production processes. This study makes a valuable contribution to the literature on malaria epidemiology and time series modeling in sub-Saharan Africa. The integration of wavelet analysis with SARIMA forecasting provides a robust framework for understanding seasonal patterns and anticipating future burdens. The emphasis on state-specific variability and synchrony offers actionable insights for malaria control programs. Thank you for choosing PLOS ONE as the venue for your work. I look forward to seeing it in print and in the scholarly conversations it will generate.

Sincerely,

Morufu Olalekan Raimi, PhD

Academic Editor

PLOS ONE
---

## [Editor Report · Acceptance letter]

PONE-D-25-59526R2

PLOS One

Dear Dr. Bakare,

I'm pleased to inform you that your manuscript has been deemed suitable for publication in PLOS One. Congratulations! Your manuscript is now being handed over to our production team.

Kind regards,

on behalf of

Prof Morufu Olalekan Raimi

Academic Editor

PLOS One